# ADASHIFT: DECORRELATION AND CONVERGENCE OF ADAPTIVE LEARNING RATE METHODS

**Zhiming Zhou**[*†]**, Qingru Zhang**[*‡]**, Guansong Lu, Hongwei Wang, Weinan Zhang, Yong Yu**
Shanghai Jiao Tong University
[†]heyohai@apex.sjtu.edu.cn,[‡]neverquit@sjtu.edu.cn

## ABSTRACT

Adam is shown not being able to converge to the optimal solution in certain cases. Researchers recently propose several algorithms to avoid the issue of non-convergence of Adam, but their efficiency turns out to be unsatisfactory in practice. In this paper, we provide a new insight into the non-convergence issue of Adam as well as other adaptive learning rate methods. We argue that there exists an inappropriate correlation between gradient $g_t$ and the second moment term $v_t$ in Adam ($t$ is the timestep), which results in that a large gradient is likely to have small step size while a small gradient may have a large step size. We demonstrate that such unbalanced step sizes are the fundamental cause of non-convergence of Adam, and we further prove that decorrelating $v_t$ and $g_t$ will lead to unbiased step size for each gradient, thus solving the non-convergence problem of Adam. Finally, we propose AdaShift, a novel adaptive learning rate method that decorrelates $v_t$ and $g_t$ by temporal shifting, i.e., using temporally shifted gradient $g_{t-n}$ to calculate $v_t$. The experiment results demonstrate that AdaShift is able to address the non-convergence issue of Adam, while still maintaining a competitive performance with Adam in terms of both training speed and generalization.

## 1 INTRODUCTION

First-order optimization algorithms with adaptive learning rate play an important role in deep learning due to their efficiency in solving large-scale optimization problems. Denote $g_t \in \mathbb{R}^n$ as the gradient of loss function $f$ with respect to its parameters $\theta \in \mathbb{R}^n$ at timestep $t$, then the general updating rule of these algorithms can be written as follows (Reddi et al., 2018):

$$\theta_{t+1} = \theta_t - \frac{\alpha_t}{\sqrt{v_t}} m_t. \tag{1}$$

In the above equation, $m_t \triangleq \phi(g_1, \ldots, g_t) \in \mathbb{R}^n$ is a function of the historical gradients; $v_t \triangleq \psi(g_1, \ldots, g_t) \in \mathbb{R}^n_+$ is an $n$-dimension vector with non-negative elements, which adapts the learning rate for the $n$ elements in $g_t$ respectively; $\alpha_t$ is the base learning rate; and $\frac{\alpha_t}{\sqrt{v_t}}$ is the adaptive step size for $m_t$.

One common choice of $\phi(g_1, \ldots, g_t)$ is the exponential moving average of the gradients used in Momentum (Qian, 1999) and Adam (Kingma & Ba, 2014), which helps alleviate gradient oscillations. The commonly-used $\psi(g_1, \ldots, g_t)$ in deep learning community is the exponential moving average of squared gradients, such as Adadelta (Zeiler, 2012), RMSProp (Tieleman & Hinton, 2012), Adam (Kingma & Ba, 2014) and Nadam (Dozat, 2016).

Adam (Kingma & Ba, 2014) is a typical adaptive learning rate method, which assembles the idea of using exponential moving average of first and second moments and bias correction. In general, Adam is robust and efficient in both dense and sparse gradient cases, and is popular in deep learning research. However, Adam is shown not being able to converge to optimal solution in certain cases. Reddi et al. (2018) point out that the key issue in the convergence proof of Adam lies in the quantity

$$\Gamma_t \triangleq \left( \frac{\sqrt{v_t}}{\alpha_t} - \frac{\sqrt{v_{t-1}}}{\alpha_{t-1}} \right), \tag{2}$$

---
[*]Equally contributed.

which is assumed to be positive, but unfortunately, such an assumption does not always hold in Adam. They provide a set of counterexamples and demonstrate that the violation of positiveness of $\Gamma_t$ will lead to undesirable convergence behavior in Adam.

Reddi et al. (2018) then propose two variants, AMSGrad and AdamNC, to address the issue by keeping $\Gamma_t$ positive. Specifically, AMSGrad defines $\hat{v}_t$ as the historical maximum of $v_t$, i.e., $\hat{v}_t = \max \{v_i\}_{i=1}^{t}$, and replaces $v_t$ with $\hat{v}_t$ to keep $v_t$ non-decreasing and therefore forces $\Gamma_t$ to be positive; while AdamNC forces $v_t$ to have "long-term memory" of past gradients and calculates $v_t$ as their average to make it stable. Though these two algorithms solve the non-convergence problem of Adam to a certain extent, they turn out to be inefficient in practice: they have to maintain a very large $v_t$ once a large gradient appears, and a large $v_t$ decreases the adaptive learning rate $\frac{\alpha_t}{\sqrt{v_t}}$ and slows down the training process.

In this paper, we provide a new insight into adaptive learning rate methods, which brings a new perspective on solving the non-convergence issue of Adam. Specifically, in Section 3, we study the counterexamples provided by Reddi et al. (2018) via analyzing the accumulated step size of each gradient $g_t$. We observe that in the common adaptive learning rate methods, a large gradient tends to have a relatively small step size, while a small gradient is likely to have a relatively large step size. We show that the unbalanced step sizes stem from the inappropriate positive correlation between $v_t$ and $g_t$, and we argue that this is the fundamental cause of the non-convergence issue of Adam.

In Section 4, we further prove that decorrelating $v_t$ and $g_t$ leads to equal and unbiased expected step size for each gradient, thus solving the non-convergence issue of Adam. We subsequently propose AdaShift, a decorrelated variant of adaptive learning rate methods, which achieves decorrelation between $v_t$ and $g_t$ by calculating $v_t$ using temporally shifted gradients. Finally, in Section 5, we study the performance of our proposed AdaShift, and demonstrate that it solves the non-convergence issue of Adam, while still maintaining a decent performance compared with Adam in terms of both training speed and generalization.

## 2 PRELIMINARIES

**Adam.** In Adam, $m_t$ and $v_t$ are defined as the exponential moving average of $g_t$ and $g_t^2$:

$$m_t = \beta_1 m_{t-1} + (1 - \beta_1)g_t \ \text{ and } \ v_t = \beta_2 v_{t-1} + (1 - \beta_2)g_t^2, \tag{3}$$

where $\beta_1 \in [0, 1)$ and $\beta_2 \in [0, 1)$ are the exponential decay rates for $m_t$ and $v_t$, respectively, with $m_0 = 0$ and $v_0 = 0$. They can also be written as:

$$m_t = (1 - \beta_1)\sum_{i=1}^{t}\beta_1^{t-i}g_i \ \text{ and } \ v_t = (1 - \beta_2)\sum_{i=1}^{t}\beta_2^{t-i}g_i^2. \tag{4}$$

To avoid the bias in the estimation of the expected value at the initial timesteps, Kingma & Ba (2014) propose to apply bias correction to $m_t$ and $v_t$. Using $m_t$ as instance, it works as follows:

$$m_t = \frac{(1 - \beta_1)\sum_{i=1}^{t}\beta_1^{t-i}g_i}{(1 - \beta_1)\sum_{i=1}^{t}\beta_1^{t-i}} = \frac{\sum_{i=1}^{t}\beta_1^{t-i}g_i}{\sum_{i=1}^{t}\beta_1^{t-i}} = \frac{(1 - \beta_1)\sum_{i=1}^{t}\beta_1^{t-i}g_i}{1 - \beta_1^t}. \tag{5}$$

**Online optimization problem.** An online optimization problem consists of a sequence of cost functions $f_1(\theta), \ldots, f_t(\theta), \ldots, f_T(\theta)$, where the optimizer predicts the parameter $\theta_t$ at each time-step $t$ and evaluate it on an unknown cost function $f_t(\theta)$. The performance of the optimizer is usually evaluated by regret $R(T) \triangleq \sum_{t=1}^{T}[f_t(\theta_t) - f_t(\theta^*)]$, which is the sum of the difference between the online prediction $f_t(\theta_t)$ and the best fixed-point parameter prediction $f_t(\theta^*)$ for all the previous steps, where $\theta^* = \arg\min_{\theta \in \vartheta} \sum_{t=1}^{T} f_t(\theta)$ is the best fixed-point parameter from a feasible set $\vartheta$.

**Counterexamples.** Reddi et al. (2018) highlight that for any fixed $\beta_1$ and $\beta_2$, there exists an online optimization problem where Adam has non-zero average regret, i.e., Adam does not converge to optimal solution . The counterexamples in the sequential version are given as follows:

$$f_t(\theta) = \begin{cases} C\theta, & \text{if t mod } d = 1; \\ -\theta, & \text{otherwise,} \end{cases} \tag{6}$$

where $C$ is a relatively large constant and $d$ is the length of an epoch. In Equation 6, most gradients of $f_t(\theta)$ with respect to $\theta$ are $-1$, but the large positive gradient $C$ at the beginning of each epoch makes the overall gradient of each epoch positive, which means that one should decrease $\theta_t$ to minimize the loss. However, according to (Reddi et al., 2018), the accumulated update of $\theta$ in Adam under some circumstance is opposite (i.e., $\theta_t$ is increased), thus Adam cannot converge in such case. Reddi et al. (2018) argue that the reason of the non-convergence of Adam lies in that the positive assumption of $\Gamma_t \triangleq (\sqrt{v_t}/\alpha_t - \sqrt{v_{t-1}}/\alpha_{t-1})$ does not always hold in Adam.

The counterexamples are also extended to stochastic cases in (Reddi et al., 2018), where a finite set of cost functions appear in a stochastic order. Compared with sequential online optimization counterexample, the stochastic version is more general and closer to the practical situation. For the simplest one dimensional case, at each timestep $t$, the function $f_t(\theta)$ is chosen as i.i.d.:

$$f_t(\theta) = \begin{cases} C\theta, & \text{with probability } p = \frac{1+\delta}{C+1}; \\ -\theta, & \text{with probability } 1-p = \frac{C-\delta}{C+1}, \end{cases} \tag{7}$$

where $\delta$ is a small positive constant that is smaller than $C$. The expected cost function of the above problem is $F(\theta) = \frac{1+\delta}{C+1}C\theta - \frac{C-\delta}{C+1}\theta = \delta\theta$, therefore, one should decrease $\theta$ to minimize the loss. Reddi et al. (2018) prove that when $C$ is large enough, the expectation of accumulated parameter update in Adam is positive and results in increasing $\theta$.

**Basic Solutions**    Reddi et al. (2018) propose maintaining the strict positiveness of $\Gamma_t$ as solution, for example, keeping $v_t$ non-decreasing or using increasing $\beta_2$. In fact, keeping $\Gamma_t$ positive is not the only way to guarantee the convergence of Adam. Another important observation is that for any fixed sequential online optimization problem with infinitely repeating epochs (e.g., Equation 6), Adam will converge as long as $\beta_1$ is large enough. Formally, we have the following theorem:

**Theorem 1** (The influence of $\beta_1$). For any fixed sequential online convex optimization problem with infinitely repeating of finite length epochs ($d$ is the length of an epoch), if $\exists G \in \mathbb{R}$ such that $\|\nabla f_t(\theta)\|_\infty \leq G$ and $\exists T \in \mathbb{N}, \exists \epsilon_2 > \epsilon_1 > 0$ such that $\epsilon_1 < \frac{\alpha_t}{\sqrt{v_t}}G^2 < \epsilon_2$ holds for all $t > T$, then, for any fixed $\beta_2 \in [0, 1)$, there exists a $\beta_1 \in [0, 1)$ such that Adam has average regret $\leq \epsilon_2$;

The intuition behind Theorem 1 is that, if $\beta_1 \to 1$, then $m_t \to \sum_{i=1}^d g_i/d$, i.e., $m_t$ approaches the average gradient of an epoch, according to Equation 5. Therefore, no matter what the adaptive learning rate $\alpha_t/\sqrt{v_t}$ is, Adam will always converge along the correct direction.

## 3    THE CAUSE OF NON-CONVERGENCE: UNBALANCED STEP SIZE

In this section, we study the non-convergence issue by analyzing the counterexamples provided by Reddi et al. (2018). We show that the fundamental problem of common adaptive learning rate methods is that: $v_t$ is positively correlated to the scale of gradient $g_t$, which results in a small step size $\alpha_t/\sqrt{v_t}$ for a large gradient, and a large step size for a small gradient. We argue that such an unbalanced step size is the cause of non-convergence.

We will first define net update factor for the analysis of the accumulated influence of each gradient $g_t$, then apply the net update factor to study the behaviors of Adam using Equation 6 as an example. The argument will be extended to the stochastic online optimization problem and general cases.

### 3.1    NET UPDATE FACTOR

When $\beta_1 \neq 0$, due to the exponential moving effect of $m_t$, the influence of $g_t$ exists in all of its following timesteps. For timestep $i$ ($i \geq t$), the weight of $g_t$ is $(1 - \beta_1)\beta_1^{i-t}$. We accordingly define a new tool for our analysis: the net update $net(g_t)$ of each gradient $g_t$, which is its accumulated influence on the entire optimization process:

$$net(g_t) \triangleq \sum_{i=t}^\infty \frac{\alpha_i}{\sqrt{v_i}}[(1-\beta_1)\beta_1^{i-t}g_t] = k(g_t) \cdot g_t, \quad \text{where } k(g_t) = \sum_{i=t}^\infty \frac{\alpha_i}{\sqrt{v_i}}(1-\beta_1)\beta_1^{i-t}, \quad (8)$$

and we call $k(g_t)$ the net update factor of $g_t$, which is the equivalent accumulated step size for gradient $g_t$. Note that $k(g_t)$ depends on $\{v_i\}_{i=t}^\infty$, and in Adam, if $\beta_1 \neq 0$, then all elements in $\{v_i\}_{i=t}^\infty$ are related to $g_t$. Therefore, $k(g_t)$ is a function of $g_t$.

It is worth noticing that in Momentum method, $v_t$ is equivalently set as 1. Therefore, we have $k(g_t) = \alpha_t$ and $net(g_t) = \alpha_t g_t$, which means that the accumulated influence of each gradient $g_t$ in Momentum is the same as vanilla SGD (Stochastic Gradient Decent). Hence, the convergence of Momentum is similar to vanilla SGD. However, in adaptive learning rate methods, $v_t$ is function over the past gradients, which makes its convergence nontrivial.

## 3.2 ANALYSIS ON ONLINE OPTIMIZATION COUNTEREXAMPLES

Note that $v_t$ exists in the definition of net update factor (Equation 8). Before further analyzing the convergence of Adam using the net update factor, we first study the pattern of $v_t$ in the sequential online optimization problem in Equation 6. Since Equation 6 is deterministic, we can derive the formula of $v_t$ as follows:

**Lemma 2.** In the sequential online optimization problem in Equation 6, denote $\beta_1, \beta_2 \in [0, 1)$ as the decay rates, $d \in \mathbb{N}$ as the length of an epoch, $n \in \mathbb{N}$ as the index of epoch, and $i \in \{1, 2, ..., d\}$ as the index of timestep in one epoch. Then the limit of $v_{nd+i}$ when $n \to \infty$ is:

$$\lim_{n \to \infty} v_{nd+i} = \frac{1 - \beta_2}{1 - \beta_2^d}(C^2 - 1)\beta_2^{i-1} + 1 . \tag{9}$$

Given the formula of $v_t$ in Equation 9, we now study the net update factor of each gradient. We start with a simple case where $\beta_1 = 0$. In this case we have

$$\lim_{n \to \infty} k(g_{nd+i}) = \lim_{n \to \infty} \frac{\alpha_t}{\sqrt{v_{nd+i}}}. \tag{10}$$

Since the limit of $v_{nd+i}$ in each epoch monotonically decreases with the increase of index $i$ according to Equation 9, the limit of $k(g_{nd+i})$ monotonically increases in each epoch. Specifically, the first gradient $g_{nd+1} = C$ in epoch $n$ represents the correct updating direction, but its influence is the smallest in this epoch. In contrast, the net update factor of the subsequent gradients $-1$ are relatively larger, though they indicate a wrong updating direction.

We further consider the general case where $\beta_1 \neq 0$. The result is presented in the following lemma:

**Lemma 3.** In the sequential online optimization problem in Equation 6, when $n \to \infty$, the limit of net update factor $k(g_{nd+i})$ of epoch $n$ satisfies: $\exists\, 1 \leq j \leq d$ such that

$$\lim_{n \to \infty} k(C) = \lim_{n \to \infty} k(g_{nd+1}) < \lim_{n \to \infty} k(g_{nd+2}) < \cdots < \lim_{n \to \infty} k(g_{nd+j}), \tag{11}$$

and

$$\lim_{n \to \infty} k(g_{nd+j}) > \lim_{n \to \infty} k(g_{nd+j+1}) > \cdots > \lim_{n \to \infty} k(g_{nd+d+1}) = \lim_{n \to \infty} k(C), \tag{12}$$

where $k(C)$ denotes the net update factor for gradient $g_i = C$.

Lemma 3 tells us that, in sequential online optimization problem in Equation 6, the net update factors are unbalanced. Specifically, the net update factor for the large gradient $C$ is the smallest in the entire epoch, while all gradients $-1$ have larger net update factors. Such unbalanced net update factors will possibly lead Adam to a wrong accumulated update direction.

Similar conclusion also holds in the stochastic online optimization problem in Equation 7. We derive the expectation of the net update factor for each gradient in the following lemma:

**Lemma 4.** In the stochastic online optimization problem in Equation 7, assuming $\alpha_t = 1$, it holds that $k(C) < k(-1)$, where $k(C)$ denote the expectation net update factor for $g_i = C$ and $k(-1)$ denote the expectation net update factor for $g_i = -1$.

Though the formulas of net update factors in the stochastic case are more complicated than those in deterministic case, the analysis is actually more easier: the gradients with the same scale share the same expected net update factor, so we only need to analyze $k(C)$ and $k(-1)$. From Lemma 4, we can see that in terms of the expectation net update factor, $k(C)$ is smaller than $k(-1)$, which means the accumulated influence of gradient $C$ is smaller than gradient $-1$.

### 3.3 Analysis on non-convergence of Adam

As we have observed in the previous section, a common characteristic of these counterexamples is that the net update factor for the gradient with large magnitude is smaller than these with small magnitude. The above observation can also be interpreted as a direct consequence of inappropriate correlation between $v_t$ and $g_t$. Recall that $v_t = \beta_2 v_{t-1} + (1 - \beta_2)g_t^2$. Assuming $v_{t-1}$ is independent of $g_t$, then: when a new gradient $g_t$ arrives, if $g_t$ is large, $v_t$ is likely to be larger; and if $g_t$ is small, $v_t$ is also likely to be smaller. If $\beta_1 = 0$, then $k(g_t) = \alpha_t/\sqrt{v_t}$. As a result, a large gradient is likely to have a small net update factor, while a small gradient is likely to have a large net update factor in Adam.

When it comes to the scenario where $\beta_1 > 0$, the arguments are actually quite similar. Given $v_t = \beta_2 v_{t-1} + (1 - \beta_2)g_t^2$. Assuming $v_{t-1}$ and $\{g_{t+i}\}_{i=1}^{\infty}$ are independent from $g_t$, then: not only does $v_t$ positively correlate with the magnitude of $g_t$, but also the entire infinite sequence $\{v_i\}_{i=t}^{\infty}$ positively correlates with the magnitude of $g_t$. Since the net update factor $k(g_t) = \sum_{i=t}^{\infty} \alpha_i/\sqrt{v_i}(1-\beta_1)\beta_1^{i-t}$ negatively correlates with each $v_i$ in $\{v_i\}_{i=t}^{\infty}$, it is thus negatively correlated with the magnitude of $g_t$. That is, $k(g_t)$ for a large gradient is likely to be smaller, while $k(g_t)$ for a small gradient is likely to be larger.

The unbalanced net update factors cause the non-convergence problem of Adam as well as all other adaptive learning rate methods where $v_t$ correlates with $g_t$. To construct a counterexample, the same pattern is that: the large gradient is along the "correct" direction, while the small gradient is along the opposite direction. Due to the fact that the accumulated influence of a large gradient is small while the accumulated influence of a small gradient is large, Adam may update parameters along the wrong direction.

Finally, we would like to emphasize that even if Adam updates parameters along the right direction in general, the unbalanced net update factors are still unfavorable since they slow down the convergence.

## 4 The proposed method: decorrelation via temporal shifting

According to the previous discussion, we conclude that the main cause of the non-convergence of Adam is the inappropriate correlation between $v_t$ and $g_t$. Currently we have two possible solutions: (1) making $v_t$ act like a constant, which declines the correlation, e.g., using a large $\beta_2$ or keep $v_t$ non-decreasing (Reddi et al., 2018); (2) using a large $\beta_1$ (Theorem 1), where the aggressive momentum term helps to mitigate the impact of unbalanced net update factors. However, neither of them solves the problem fundamentally.

The dilemma caused by $v_t$ enforces us to rethink its role. In adaptive learning rate methods, $v_t$ plays the role of estimating the second moments of gradients, which reflects the scale of gradient on average. With the adaptive learning rate $\alpha_t/\sqrt{v_t}$, the update step of $g_t$ is scaled down by $\sqrt{v_t}$ and achieves rescaling invariance with respect to the scale of $g_t$, which is practically useful to make the training process easy to control and the training system robust. However, the current scheme of $v_t$, i.e., $v_t = \beta_2 v_{t-1} + (1 - \beta_2)g_t^2$, brings a positive correlation between $v_t$ and $g_t$, which results in reducing the effect of large gradients and increasing the effect of small gradients, and finally causes the non-convergence problem. Therefore, the key is to let $v_t$ be a quantity that reflects the scale of the gradients, while at the same time, be decorrelated with current gradient $g_t$. Formally, we have the following theorem:

**Theorem 5** (Decorrelation leads to convergence). For any fixed online optimization problem with infinitely repeating of a finite set of cost functions $\{f_1(\theta), \dots, f_t(\theta), \dots f_n(\theta)\}$, assuming $\beta_1 = 0$ and $\alpha_t$ is fixed, we have, if $v_t$ follows a fixed distribution and is independent of the current gradient $g_t$, then the expected net update factor for each gradient is identical.

Let $P_v$ denote the distribution of $v_t$. In the infinitely repeating online optimization scheme, the expectation of net update factor for each gradient $g_t$ is

$$\mathbb{E}[k(g_t)] = \sum_{i=t}^{\infty} \mathbb{E}_{v_i \sim P_v}[\frac{\alpha_i}{\sqrt{v_i}}(1 - \beta_1)\beta_1^{i-t}]. \tag{13}$$

Given $P_v$ is independent of $g_t$, the expectation of the net update factor $\mathbb{E}[k(g_t)]$ is independent of $g_t$ and remains the same for different gradients. With the expected net update factor being a fixed constant, the convergence of the adaptive learning rate method reduces to vanilla SGD.

Momentum (Qian, 1999) can be viewed as setting $v_t$ as a constant, which makes $v_t$ and $g_t$ independent. Furthermore, in our view, using an increasing $\beta_2$ (AdamNC) or keeping $\hat{v}_t$ as the largest $v_t$ (AMSGrad) is also to make $v_t$ almost fixed. However, fixing $v_t$ is not a desirable solution, because it damages the adaptability of Adam with respect to the adapting of step size.

We next introduce the proposed solution to make $v_t$ independent of $g_t$, which is based on temporal independent assumption among gradients. We first introduce the idea of temporal decorrelation, then extend our solution to make use of the spatial information of gradients. Finally, we incorporate first moment estimation. The pseudo code of the proposed algorithm is presented as follows.

---

**Algorithm 1** AdaShift: Temporal Shifting with Block-wise Spatial Operation

---

**Input:** $n, \beta_1, \beta_2, \phi, \theta_0, \{f_t(\theta)\}_{t=1}^T, \{\alpha_t\}_{t=1}^T, \{g_{-t}\}_{t=0}^{n-1}$,
1: set $v_0 = 0$
2: **for** $t = 1$ **to** $T$ **do**
3:     $g_t = \nabla f_t(\theta_t)$
4:     $m_t = \sum_{i=0}^{n-1} \beta_1^i g_{t-i} / \sum_{i=0}^{n-1} \beta_1^i$
5:     **for** $i = 1$ **to** $M$ **do**
6:        $v_t[i] = \beta_2 v_{t-1}[i] + (1 - \beta_2)\phi(g_{t-n}^2[i])$
7:        $\theta_t[i] = \theta_{t-1}[i] - \alpha_t / \sqrt{v_t[i]} \cdot m_t[i]$
8:     **end for**
9: **end for**
10: // We ignore the bias-correction, epsilon and other misc for the sake of clarity

---

### 4.1 TEMPORAL DECORRELATION

In practical setting, $f_t(\theta)$ usually involves different mini-batches $x_t$, i.e., $f_t(\theta) = f(\theta; x_t)$. Given the randomness of mini-batch, we assume that the mini-batch $x_t$ is independent of each other and further assume that $f(\theta; x)$ keeps unchanged over time, then the gradient $g_t = \nabla f(\theta; x_t)$ of each mini-batch is independent of each other.

Therefore, we could change the update rule for $v_t$ to involve $g_{t-n}$ instead of $g_t$, which makes $v_t$ and $g_t$ temporally shifted and hence decorrelated:

$$v_t = \beta_2 v_{t-1} + (1 - \beta_2) g_{t-n}^2. \tag{14}$$

Note that in the sequential online optimization problem, the assumption "$g_t$ is independent of each other" does not hold. However, in the stochastic online optimization problem and practical neural network settings, our assumption generally holds.

### 4.2 MAKING USE OF THE SPATIAL ELEMENTS OF PREVIOUS TIMESTEPS

Most optimization schemes involve a great many parameters. The dimension of $\theta$ is high, thus $g_t$ and $v_t$ are also of high dimension. However, $v_t$ is element-wisely computed in Equation 14. Specifically, we only use the $i$-th dimension of $g_{t-n}$ to calculate the $i$-th dimension of $v_t$. In other words, it only makes use of the independence between $g_{t-n}[i]$ and $g_t[i]$, where $g_t[i]$ denotes the $i$-th element of $g_t$. Actually, in the case of high-dimensional $g_t$ and $v_t$, we can further assume that **all elements of gradient** $g_{t-n}$ **at previous timesteps** are independent with the $i$-th dimension of $g_t$. Therefore, all elements in $g_{t-n}$ can be used to compute $v_t$ without introducing correlation. To this end, we propose introducing a function $\phi$ over all elements of $g_{t-n}^2$, i.e.,

$$v_t = \beta_2 v_{t-1} + (1 - \beta_2)\phi(g_{t-n}^2). \tag{15}$$

For easy reference, we name the elements of $g_{t-n}$ other than $g_{t-n}[i]$ as the spatial elements of $g_{t-n}$ and name $\phi$ the spatial function or spatial operation. There is no restriction on the choice of $\phi$, and we use $\phi(x) = \max_i x[i]$ for most of our experiments, which is shown to be a good choice. The $\max_i x[i]$ operation has a side effect that turns the adaptive learning rate $v_t$ into a shared scalar.

An important thing here is that, we no longer interpret $v_t$ as the second moment of $g_t$. It is merely a random variable that is independent of $g_t$, while at the same time, reflects the overall gradient scale. We leave further investigations on $\phi$ as future work.

## 4.3 Block-wise adaptive learning rate SGD

In practical setting, e.g., deep neural network, $\theta$ usually consists of many parameter blocks, e.g., the weight and bias for each layer. In deep neural network, the gradient scales (i.e., the variance) for different layers tend to be different (Glorot & Bengio, 2010; He et al., 2015). Different gradient scales make it hard to find a learning rate that is suitable for all layers, when using SGD and Momentum methods. In traditional adaptive learning rate methods, they apply element-wise rescaling for each gradient dimension, which achieves rescaling-invariance and somehow solves the above problem. However, Adam sometimes does not generalize better than SGD (Wilson et al., 2017; Keskar & Socher, 2017), which might relate to the excessive learning rate adaptation in Adam.

In our temporal decorrelation with spatial operation scheme, we can solve the "different gradient scales" issue more naturally, by applying $\phi$ block-wisely and outputs a shared adaptive learning rate scalar $v_t[i]$ for each block:

$$v_t[i] = \beta_2 v_{t-1}[i] + (1 - \beta_2)\phi(g_{t-n}^2[i]).\qquad(16)$$

It makes the algorithm work like an adaptive learning rate SGD, where each block has an adaptive learning rate $\alpha_t/\sqrt{v_t[i]}$ while the relative gradient scale among in-block elements keep unchanged. As illustrated in Algorithm 1, the parameters $\theta_t$ including the related $g_t$ and $v_t$ are divided into $M$ blocks. Every block contains the parameters of the same type or same layer in neural network.

## 4.4 Incorporating first moment estimation: moving averaging windows

First moment estimation, i.e., defining $m_t$ as a moving average of $g_t$, is an important technique of modern first order optimization algorithms, which alleviates mini-batch oscillations. In this section, we extend our algorithm to incorporate first moment estimation.

We have argued that $v_t$ needs to be decorrelated with $g_t$. Analogously, when introducing the first moment estimation, we need to make $v_t$ and $m_t$ independent to make the expected net update factor unbiased. Based on our assumption of temporal independence, we further keep out the latest $n$ gradients $\{g_{t-i}\}_{i=0}^{n-1}$, and update $v_t$ and $m_t$ via

$$v_t = \beta_2 v_{t-1} + (1 - \beta_2)\phi(g_{t-n}^2) \text{ and } m_t = \frac{\sum_{i=0}^{n-1} \beta_1^i g_{t-i}}{\sum_{i=0}^{n-1} \beta_1^i}.\qquad(17)$$

In Equation 17, $\beta_1 \in [0, 1]$ plays the role of decay rate for temporal elements. It can be viewed as a truncated version of exponential moving average that only applied to the latest few elements. Since we use truncating, it is feasible to use large $\beta_1$ without taking the risk of using too old gradients. In the extreme case where $\beta_1 = 1$, it becomes vanilla averaging.

The pseudo code of the algorithm that unifies all proposed techniques is presented in Algorithm 1 and a more detailed version can be found in the Appendix. It has the following parameters: spatial operation $\phi$, $n \in \mathbb{N}^+$, $\beta_1 \in [0, 1]$, $\beta_2 \in [0, 1)$ and $\alpha_t$.

**Summary** The key difference between Adam and the proposed method is that the latter temporally shifts the gradient $g_t$ for $n$-step, i.e., using $g_{t-n}$ for calculating $v_t$ and using the kept-out $n$ gradients to evaluate $m_t$ (Equation 17), which makes $v_t$ and $m_t$ decorrelated and consequently solves the non-convergence issue. In addition, based on our new perspective on adaptive learning rate methods, $v_t$ is not necessarily the second moment and it is valid to further involve the calculation of $v_t$ with the spatial elements of previous gradients. We thus proposed to introduce the spatial operation $\phi$ that outputs a shared scalar for each block. The resulting algorithm turns out to be closely related to SGD, where each block has an overall adaptive learning rate and the relative gradient scale in each block is maintained. We name the proposed method that makes use of temporal-shifting to decorrelated $v_t$ and $m_t$ **AdaShift**, which means "ADAptive learning rate method with temporal SHIFTing".

## 5 EXPERIMENTS

In this section, we empirically study the proposed method and compare them with Adam, AMSGrad and SGD, on various tasks in terms of training performance and generalization. Without additional declaration, the reported result for each algorithm is the best we have found via parameter grid search. The anonymous code is provided at http://bit.ly/2NDXX6x.

### 5.1 ONLINE OPTIMIZATION COUNTEREXAMPLES

Firstly, we verify our analysis on the stochastic online optimization problem in Equation 7, where we set $C = 101$ and $\delta = 0.02$. We compare Adam, AMSGrad and AdaShift in this experiment. For fair comparison, we set $\alpha = 0.001$, $\beta_1 = 0$ and $\beta_2 = 0.999$ for all these methods. The results are shown in Figure 1a. We can see that Adam tends to increase $\theta$, that is, the accumulate update of $\theta$ in Adam is along the wrong direction, while AMSGrad and AdaShift update $\theta$ in the correct direction. Furthermore, given the same learning rate, AdaShift decreases $\theta$ faster than AMSGrad, which validates our argument that AMSGrad has a relatively higher $v_t$ that slows down the training.

In this experiment, we also verify Theorem 1. As shown in Figure 1b, Adam is also able to converge to the correct direction with a sufficiently large $\beta_1$ and $\beta_2$. Note that (1) AdaShift still converges with the fastest speed; (2) a small $\beta_1$ (e.g., $\beta_1 = 0.9$, the light-blue line in Figure 1b) does not make Adam converge to the correct direction. We do not conduct the experiments on the sequential online optimization problem in Equation 6, because it does not fit our temporal independence assumption. To make it converge, one can use a large $\beta_1$ or $\beta_2$, or set $v_t$ as a constant.

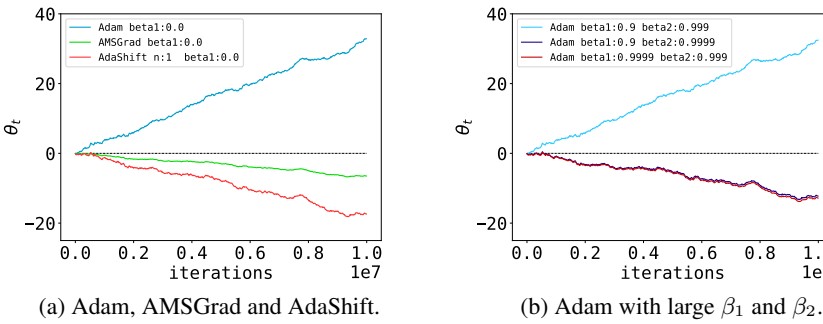

(a) Adam, AMSGrad and AdaShift.          (b) Adam with large $\beta_1$ and $\beta_2$.

Figure 1: Experiments on stochastic counterexample.

### 5.2 LOGISTIC REGRESSION AND MULTILAYER PERCEPTRON ON MNIST

We further compare the proposed method with Adam, AMSGrad and SGD by using Logistic Regression and Multilayer Perceptron on MNIST, where the Multilayer Perceptron has two hidden layers and each has 256 hidden units with no internal activation. The results are shown in Figure 2 and Figure 3, respectively. We find that in Logistic Regression, these learning algorithms achieve very similar final results in terms of both training speed and generalization. In Multilayer Perceptron, we compare Adam, AMSGrad and AdaShift with reduce-max spatial operation (max-AdaShift) and without spatial operation (non-AdaShift). We observe that max-AdaShift achieves the lowest training loss, while non-AdaShift has mild training loss oscillation and at the same time achieves better generalization. The worse generalization of max-AdaShift may be due to overfitting in this task, and the better generalization of non-AdaShift may stem from the regularization effect of its relatively unstable step size.

### 5.3 DENSENET AND RESNET ON CIFAR-10

ResNet (He et al., 2016) and DenseNet (Huang et al., 2017) are two typical modern neural networks, which are efficient and widely-used. We test our algorithm with ResNet and DenseNet on CIFAR-10 datasets. We use a 18-layer ResNet and 100-layer DenseNet in our experiments. We plot the best results of Adam, AMSGrad and AdaShift in Figure 4 and Figure 5 for ResNet and DenseNet, respectively. We can see that AMSGrad is relatively worse in terms of both training speed and generalization. Adam and AdaShift share competitive results, while AdaShift is generally slightly better, especially the test accuracy of ResNet and the training loss of DenseNet.

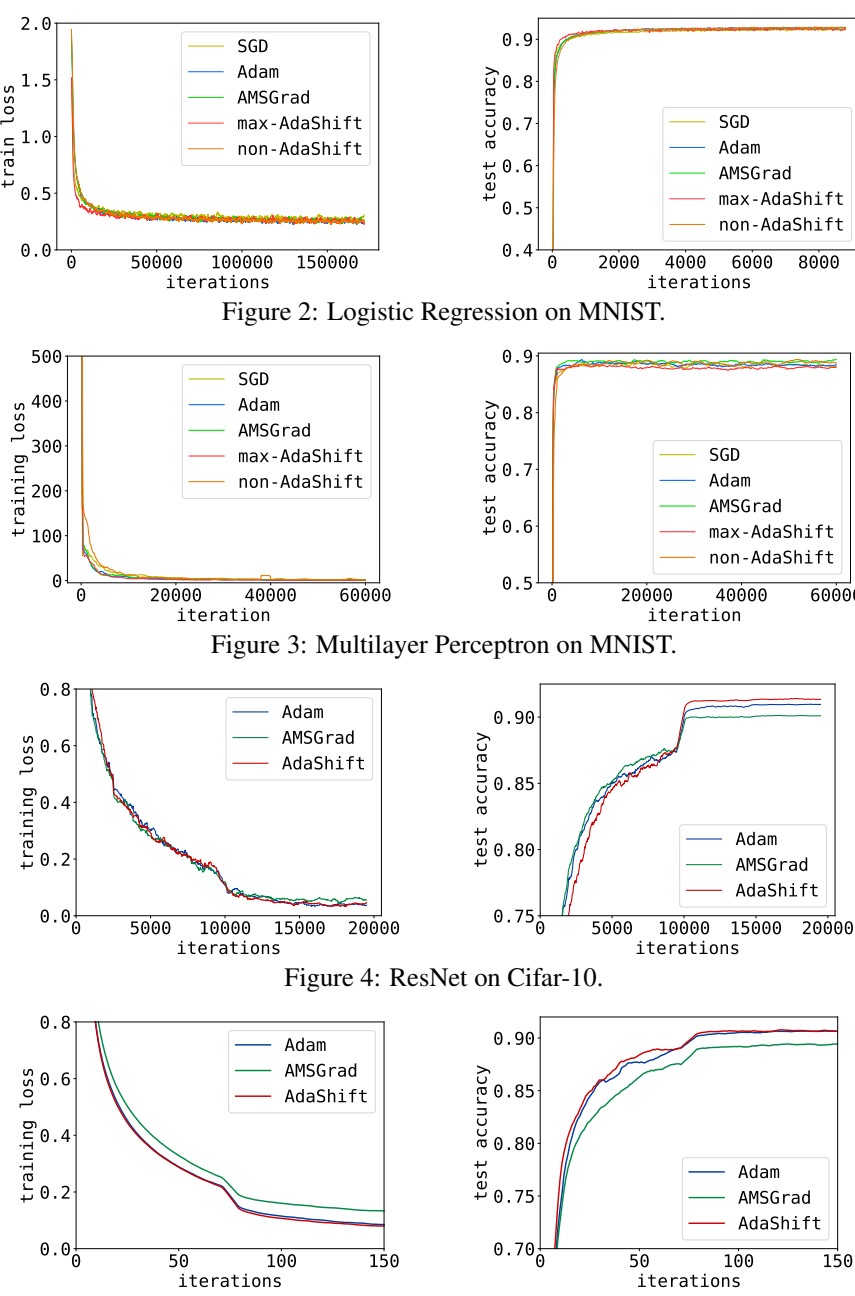

Figure 2: Logistic Regression on MNIST.

Figure 3: Multilayer Perceptron on MNIST.

Figure 4: ResNet on Cifar-10.

Figure 5: DenseNet on Cifar-10.

### 5.4 DenseNet with Tiny-ImageNet

We further increase the complexity of dataset, switching from CIFAR-10 to Tiny-ImageNet, and compare the performance of Adam, AMSGrad and AdaShift with DenseNet. The results are shown in Figure 6, from which we can see that the training curves of Adam and AdaShift are basically overlapped, but AdaShift achieves higher test accuracy than Adam. AMSGrad has relatively higher training loss, and its test accuracy is relatively lower at the initial stage.

### 5.5 Generative model and Recurrent model

We also test our algorithm on the training of generative model and recurrent model. We choose WGAN-GP (Gulrajani et al., 2017) that involves Lipschitz continuity condition (which is hard to optimize), and Neural Machine Translation (NMT) (Luong et al., 2017) that involves typical recurrent unit LSTM, respectively. In Figure 7a, we compare the performance of Adam, AMSGrad

and AdaShift in the training of WGAN-GP discriminator, given a fixed generator. We notice that AdaShift is significantly better than Adam, while the performance of AMSGrad is relatively unsatisfactory. The test performance in terms of BLEU of NMT is shown in Figure 7b, where AdaShift achieves a higher BLEU than Adam and AMSGrad.

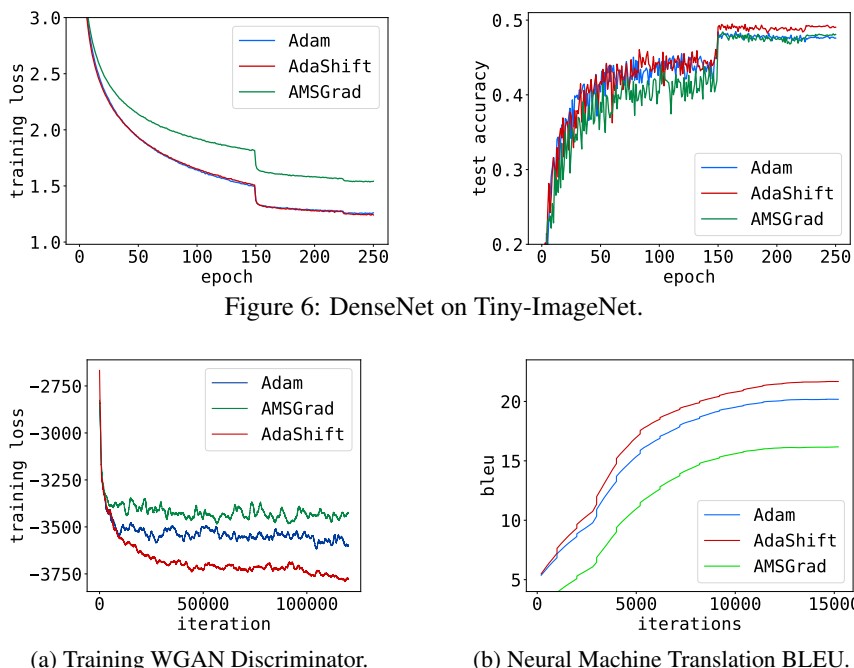

Figure 6: DenseNet on Tiny-ImageNet.

(a) Training WGAN Discriminator.     (b) Neural Machine Translation BLEU.

Figure 7: Generative and Recurrent model.

## 6   CONCLUSION

In this paper, we study the non-convergence issue of adaptive learning rate methods from the perspective of the equivalent accumulated step size of each gradient, i.e., the net update factor defined in this paper. We show that there exists an inappropriate correlation between $v_t$ and $g_t$, which leads to unbalanced net update factor for each gradient. We demonstrate that such unbalanced step sizes are the fundamental cause of non-convergence of Adam, and we further prove that decorrelating $v_t$ and $g_t$ will lead to unbiased expected step size for each gradient, thus solving the non-convergence problem of Adam. Finally, we propose AdaShift, a novel adaptive learning rate method that decorrelates $v_t$ and $g_t$ via calculating $v_t$ using temporally shifted gradient $g_{t-n}$.

In addition, based on our new perspective on adaptive learning rate methods, $v_t$ is no longer necessarily the second moment of $g_t$, but a random variable that is independent of $g_t$ and reflects the overall gradient scale. Thus, it is valid to calculate $v_t$ with the spatial elements of previous gradients. We further found that when the spatial operation $\phi$ outputs a shared scalar for each block, the resulting algorithm turns out to be closely related to SGD, where each block has an overall adaptive learning rate and the relative gradient scale in each block is maintained. The experiment results demonstrate that AdaShift is able to solve the non-convergence issue of Adam. In the meantime, AdaShift achieves competitive and even better training and testing performance when compared with Adam.

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

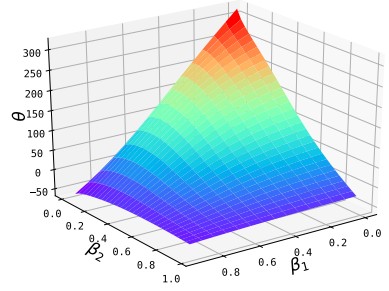

(a) Final result of $\theta$ for sequential problem after 2000 updates, varied with $\beta_1$ and $\beta_2$.

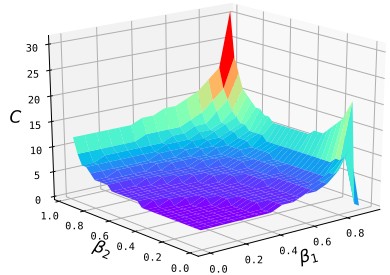

(b) Critical value of $C$ with varying $\beta_1$ and $\beta_2$ under the sequential optimization setting.

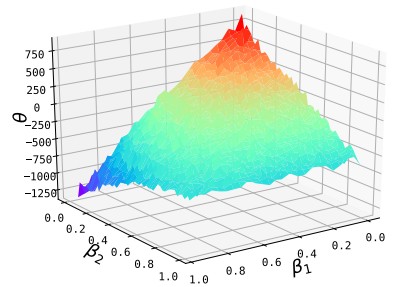

(c) Final result of $\theta$ for stochastic problem after 2000 updates, varied with $\beta_1$ and $\beta_2$.

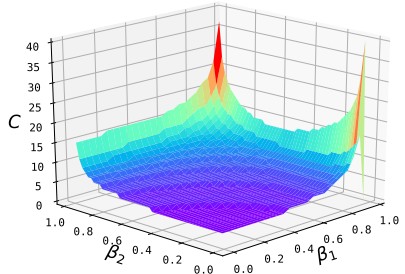

(d) Critical value of $C$ with varying $\beta_1$ and $\beta_2$ under the stochastic optimization setting.

Figure 8: Both $\beta_1$ and $\beta_2$ influence the direction and speed of optimization in Adam. Critical value of $C_t$, at which Adam gets into non-convergence, increases as $\beta_1$ and $\beta_2$ getting large. Leftmost two for the sequential online optimization problem and rightmost two for stochastic online problem.

## A   THE RELATION AMONG $\beta_1$, $\beta_2$ AND $C$

To provide an intuitive impression on the relation among $C, d, \beta_1, \beta_2$ and the convergence of Adam, we let $C = d = 6$, initialize $\theta_1 = 0$, vary $\beta_1$ and $\beta_2$ among $[0, 1)$ and let Adam go through 2000 timesteps (iterations). The final result of $\theta$ is shown in Figure 8a. It suggests that for a fixed sequential online optimization problem, both of $\beta_1$ and $\beta_2$ determine the direction and speed of Adam optimization process. Furthermore, we also study the threshold point of $C$ and $d$, under which Adam will change to the incorrect direction, for each fixed $\beta_1$ and $\beta_2$ that vary among $[0, 1)$. To simplify the experiments, we keep $d = C$ such that the overall gradient of each epoch being $+1$. The result is shown in Figure 8b, which suggests, at the condition of larger $\beta_1$ or larger $\beta_2$, it needs a larger $C$ to make Adam stride on the opposite direction. In other words, large $\beta_1$ and $\beta_2$ will make the non-convergence rare to happen.

We also conduct the experiment in the stochastic problem to analyze the relation among $C, \beta_1, \beta_2$ and the convergence behavior of Adam. Results are shown in the Figure 8c and Figure 8d and the observations are similar to the previous: larger $C$ will cause non-convergence more easily and a larger $\beta_1$ or $\beta_2$ somehow help to resolve non-convergence issue. In this experiment, we set $\delta = 1$.

**Lemma 6** (Critical condition). In the sequential online optimization problem Equation 6, let $\alpha_t$ being fixed, define $\mathcal{S}(\beta_1, \beta_2, C, d)$ to be the sum of the limits of step updates in a $d$-step epoch:

$$\mathcal{S}(\beta_1, \beta_2, C) \triangleq \sum_{i=1}^{d} \lim_{nd \to \infty} \frac{m_{nd+i}}{\sqrt{v_{nd+i}}} \ . \tag{18}$$

Let $\mathcal{S}(\beta_1, \beta_2, C) = 0$, assuming $\beta_2$ and $C$ are large enough such that $v_t \gg 1$, we get the equation:

$$C + 1 = \frac{(1 - \beta_1^d)(\sqrt{\beta_2^d} - \beta_1^d)(1 - \sqrt{\beta_2})}{(1 - \beta_1)(\sqrt{\beta_2} - \beta_1)(1 - \sqrt{\beta_2^d})} \ . \tag{19}$$

Equation 19, though being quite complex, tells that both $\beta_1$ and $\beta_2$ are closely related to the counterexamples, and there exists a critical condition among these parameters.

## B  THE ADASHIFT PSEUDO CODE

**Algorithm 2** AdaShift: We use a first-in-first-out queue $Q$ to denote the averaging window with the length of $n$. $Push(Q, g_t)$ denotes pushing vector $g_t$ to the tail of $Q$, while $Pop(Q)$ pops and returns the head vector of $Q$. And $W$ is the weight vector calculated via $\beta_1$.

**Input:** $n$, $\beta_1$, $\beta_2$, $\phi$, $\epsilon$, $\theta_0$, $\{f_t(\theta)\}_{t=1}^T$, $\{\alpha_t\}_{t=1}^T$
1: set $v_0 = 0$, $p_0 = 1$
2: $W = [\beta_1^{n-1}, \beta_1^{n-2}, \ldots, \beta_1, 1]/\sum_{i=0}^{n-1} \beta_1^n$
3: **for** $t = 1$ **to** $T$ **do**
4:    $g_t = \nabla f_t(\theta_t)$
5:    **if** $t \leq n$ **then**
6:      $Push(Q, g_t)$
7:    **else**
8:      $g_{t-n} = Pop(Q)$
9:      $Push(Q, g_t)$
10:     $m_t = W \cdot Q$
11:     $p_t = p_{t-1}\beta_2$
12:     **for** $i = 1$ **to** $M$ **do**
13:       $v_t[i] = \beta_2 v_{t-1}[i] + (1 - \beta_2)\phi(g_{t-n}^2[i])$
14:       $\theta_t[i] = \theta_{t-1}[i] - \alpha_t/(\sqrt{v_t[i]/(1-p_t)} + \epsilon) \cdot m_t[i]$
15:     **end for**
16:    **end if**
17: **end for**

We provided the anonymous code where a Tensorflow implementation of this algorithm is available.

## C  CORRELATION BETWEEN $g_t$ AND $v_t$

In order to verify the correlation between $g_t$ and $v_t$ in Adam and AdaShift, we conduct experiments to calculate the correlation coefficient between $g_t$ and $v_t$. We train the Multilayer Perceptron on MNIST until converge and gather the gradient of the second hidden layer of each step. Based on these data, we calculate $v_t$ and the correlation coefficient between $g_t[i]$ and $g_{t-n}[i]$, between $g_t[i]$ and $g_{t-n}[j]$ and between $g_t[i]$ and $v_t[i]$ of the last 10 epochs using the Pearson correlation coefficient, which is formulated as follows:

$$\rho = \frac{\sum_{i=1}^n (X_i - \bar{X})(Y_i - \bar{Y})}{\sqrt{\sum_{i=1}^n (X_i - \bar{X})^2}\sqrt{\sum_{i=1}^n (Y_i - \bar{Y})^2}}.$$

To verify the temporal correlation between $g_t[i]$ and $g_{t-n}[i]$, we range $n$ from 1 to 10 and calculate the average temporal correlation coefficient of all variables $i$. Results are shown in Table 1.

Table 1: Temporal correlation coefficient between $g_t[i]$ and $g_{t-n}[i]$.

| $n$ | 1 | 2 | 3 | 4 | 5 |
|---|---|---|---|---|---|
| $\rho$ | -0.000368929 | -0.000989286 | -0.001540511 | -0.00116966 | -0.001613395 |
| $n$ | 6 | 7 | 8 | 9 | 10 |
| $\rho$ | -0.001211721 | 0.000357474 | -0.00082293 | -0.001755237 | -0.001267641 |

To verify the spatial correlation between $g_t[i]$ and $g_{t-n}[j]$, we again range $n$ from 1 to 10 and randomly sample some pairs of $i$ and $j$ and calculate the average spatial correlation coefficient of all the selected pairs. Results are shown in Table 2.

To verify the correlation between $g_t[i]$ and $v_t[i]$ within Adam, we calculate $v_t$ and the average correlation coefficient between $g_t^2$ and $v_t$ of all variables $i$. The result is **0.435885276**.

To verify the correlation between $g_{t-n}[i]$ and $v_t[i]$ within non-AdaShift and between $g_{t-n}[i]$ and $v_t$ within max-AdaShift, we range the keep number $n$ from 1 to 10 to calculate $v_t$ and the average correlation coefficient of all variables $i$. The result is shown in Table 3 and Table 4.

Table 2: Spatial correlation coefficient between $g_t[i]$ and $g_{t-n}[j]$.

| $n$ | 1 | 2 | 3 | 4 | 5 |
|---|---|---|---|---|---|
| $\rho$ | -0.000609471 | -0.001948853 | -0.001426661 | 0.000904615 | 0.000329359 |
| $n$ | 6 | 7 | 8 | 9 | 10 |
| $\rho$ | 0.000971337 | -0.000644563 | -0.00137805 | -0.001147973 | -0.000592037 |

Table 3: Correlation coefficient between $g_{t-n}^2[i]$ and $v_t[i]$ in non-AdaShift.

| $n$ | 1 | 2 | 3 | 4 | 5 |
|---|---|---|---|---|---|
| $\rho$ | -0.010897023 | -0.010952548 | -0.010890854 | -0.010853069 | -0.010810747 |
| $n$ | 6 | 7 | 8 | 9 | 10 |
| $\rho$ | -0.010777789 | -0.01075946 | -0.010739279 | -0.010728553 | -0.010720019 |

Table 4: Correlation coefficient between $g_{t-n}^2[i]$ and $v_t$ in max-AdaShift.

| $n$ | 1 | 2 | 3 | 4 | 5 |
|---|---|---|---|---|---|
| $\rho$ | -0.000706289 | -0.000794959 | -0.00076306 | -0.000712474 | -0.000668459 |
| $n$ | 6 | 7 | 8 | 9 | 10 |
| $\rho$ | -0.000623162 | -0.000566573 | -0.000542046 | -0.000598015 | -0.000592707 |

## D  PROOF OF THEOREM 1

*Proof.*

With bias correction, the formulation of $m_t$ is written as follows

$$m_t = \frac{(1-\beta_1)\sum_{i=1}^t \beta_1^{t-i} g_i}{(1-\beta_1)\sum_{i=1}^t \beta_1^{t-i}} = \frac{\sum_{i=1}^t \beta_1^{t-i} g_i}{\sum_{i=1}^t \beta_1^{t-i}}. \tag{20}$$

According to L'Hospitals rule, we can draw the following:

$$\lim_{\beta_1 \to 1} \sum_{i=1}^t \beta_1^{t-i} = \lim_{\beta_1 \to 1} \frac{1-\beta_1^t}{1-\beta_1} = t.$$

Thus,

$$\lim_{\beta_1 \to 1} m_t = \frac{\sum_{i=1}^t g_i}{t}.$$

According to the definition of limitation, let $g^* = \frac{\sum_{i=1}^t g_i}{t}$, we have, $\forall \epsilon > 0, \exists \beta_1 \in (0,1)$, such that

$$\|m_t - g^*\|_\infty < \epsilon.$$

We set $\epsilon$ to be $\left|\frac{g^*}{2}\right|$, then for each dimension of $m_t$, i.e. $m_t[i]$,

$$\frac{g^*[i]}{2} \le m_t[i] \le \frac{3g^*[i]}{2}$$

So, $m_t$ shares the same sign with $g^*$ in every dimension.

Given it is a convex optimization problem, let the optimal parameter be $\theta^*$, and the maximum step size is $\frac{\alpha_t}{\sqrt{v_t}}G$ that holds $\epsilon_1/G < \frac{\alpha_t}{\sqrt{v_t}}G < \epsilon_2/G$, we have,

$$\lim_{t \to \infty} \|\theta_t - \theta^*\|_\infty < \epsilon_2/G. \tag{21}$$

Given $\|\nabla f_t(\theta)\|_\infty \le G$, we have $f_t(\theta) - f_t(\theta^*) < \epsilon_2$, which implies the average regret

$$R(T)/T = \sum_{t=1}^T [f_t(\theta_t) - f_t(\theta^*)]/T < \epsilon_2. \tag{22}$$

$\square$

## E   PROOF OF LEMMA 2

*Proof.* Let $\beta_1, \beta_2 \in [0, 1)$, $d \in \mathbb{N}$, $1 \le i \le d$ and $i \in \mathbb{N}$.

$$
\begin{aligned}
m_{nd+i} &= (1 - \beta_1) \sum_{j=1}^{nd+i} \beta_1^{nd+i-j} g_j \\
&= (1 - \beta_1) \left[ (C + 1) \sum_{j=0}^{n} \beta_1^{jd+i-1} - \sum_{j=0}^{nd+i-1} \beta_1^{j} \right] \\
&= (1 - \beta_1) \left[ \frac{1 - \beta_1^{(n+1)d}}{1 - \beta_1^d} \beta_1^{i-1} (C + 1) - \frac{1 - \beta_1^{nd+i}}{1 - \beta_1} \right] \\
&= \frac{1 - \beta_1^{(n+1)d}}{1 - \beta_1^d} (1 - \beta_1) \beta_1^{i-1} (C + 1) - (1 - \beta_1^{nd+i})
\end{aligned}
$$

For a fixed $d$, as $n$ approach infinity, we get the limit of $m_{nd+i}$ as:

$$
\lim_{nd \to \infty} m_{nd+i} = \frac{1 - \beta_1}{1 - \beta_1^d} (C + 1) \beta_1^{i-1} - 1
$$

Similarly, for $v_{nd+i}$:

$$
\begin{aligned}
v_{nd+i} &= (1 - \beta_2) \sum_{j=1}^{nd+i} \beta_2^{nd+i-j} g_j^2 \\
&= (1 - \beta_2) \left[ (C^2 - 1) \sum_{j=0}^{n} \beta_2^{jd+i-1} + \sum_{j=0}^{nd+i-1} \beta_2^{j} \right] \\
&= (1 - \beta_2) \left[ \frac{1 - \beta_2^{(n+1)d}}{1 - \beta_2^d} \beta_2^{i-1} (C^2 - 1) + \frac{1 - \beta_2^{nd+i}}{1 - \beta_2} \right] \\
&= \frac{1 - \beta_2^{(n+1)d}}{1 - \beta_2^d} (1 - \beta_2) \beta_2^{i-1} (C + 1) - (1 - \beta_2^{nd+i})
\end{aligned}
$$

For a fixed $d$, as $n$ approach infinity, we get the limit of $v_{nd+i}$ as:

$$
\lim_{nd \to \infty} v_{nd+i} = \frac{1 - \beta_2}{1 - \beta_2^d} (C^2 - 1) \beta_2^{i-1} + 1 \ .
$$

$\square$

## F   PROOF OF LEMMA 3

*Proof.* First, we define $\widetilde{V}_i$ as:

$$
\widetilde{V}_i = \lim_{nd \to \infty} \frac{1}{\sqrt{v_{nd+i}}} = \frac{1}{\sqrt{\frac{1 - \beta_2}{1 - \beta_2^d} (C^2 - 1) \beta_2^{(i-1)} + 1}}
$$

where $1 \le i \le d$ and $i \in \mathbb{N}$. And $\widetilde{V}_i$ has a period of $d$. Let $t^{'} = t - nd$, then we can draw:

$$\lim_{nd\to\infty} k(g_{nd+i}) = \sum_{t=nd+i}^{\infty} \frac{(1-\beta_1)\beta_1^{t-nd-i}}{\sqrt{\frac{1-\beta_2}{1-\beta_2^d}(C^2-1)\beta_2^{(t-1)\,\%\,d}+1}}$$

$$= \sum_{t'=i}^{\infty} \frac{(1-\beta_1)\beta_1^{t'-i}}{\sqrt{\frac{1-\beta_2}{1-\beta_2^d}(C^2-1)\beta_2^{(t'-1)\,\%\,d}+1}}$$

$$= \sum_{l=1}^{\infty} \sum_{j''=(l-1)d+i}^{ld+i-1} (1-\beta_1)\beta_1^{j''-i}\cdot\widetilde{V}_{j''}$$

$$= \sum_{l=1}^{\infty} \beta_1^{(l-1)d} \sum_{j'=i}^{i+d-1} (1-\beta_1)\beta_1^{j'-i}\cdot\widetilde{V}_{j'}$$

$$= \sum_{l=1}^{\infty} \beta_1^{(l-1)d} \sum_{j=0}^{d-1} (1-\beta_1)\beta_1^{j}\cdot\widetilde{V}_{j+i}$$

$$= \sum_{l=1}^{\infty} \beta_1^{(l-1)d} \left[ \beta_1\sum_{j=0}^{d-1} (1-\beta_1)\beta_1^{j}\cdot\widetilde{V}_{j+i+1} + (1-\beta_1)(1-\beta_1^d)\cdot\widetilde{V}_i \right]$$

$$= \beta_1\cdot\lim_{nd\to\infty} k(g_{nd+i+1}) + \sum_{l=1}^{\infty} \beta_1^{(l-1)d}(1-\beta_1)(1-\beta_1^d)\cdot\widetilde{V}_i$$

Thus, we can get the forward difference of $k(g_{nd+i})$ as:

$$\lim_{nd\to\infty} k(g_{nd+i+1}) - \lim_{nd\to\infty} k(g_{nd+i}) = \sum_{l=1}^{\infty} \beta_1^{(l-1)d} \left[ (1-\beta_1)^2\sum_{j=0}^{\infty}\beta_1^{j}\cdot\widetilde{V}_{j+i+1} + (1-\beta_1)^2\sum_{j=0}^{\infty}\beta_1^{j}\cdot\widetilde{V}_i \right]$$

$$= (1-\beta_1)^2\sum_{l=1}^{\infty} \beta_1^{(l-1)d} \sum_{j=0}^{d-1}\beta_1^{j}\cdot\left[\widetilde{V}_{j+i+1} - \widetilde{V}_i\right]$$

$\widetilde{V}_{nd+1i}$ monotonically increases within one period, when $1 \leq i \leq d$ and $i \in \mathbb{N}$. And the weigh $\beta_1^j$ for every difference term $\left[\widetilde{V}_{j+i+1} - \widetilde{V}_i\right]$ is fixed when $i$ varies. Thus, the weighted summation $\sum_{j=0}^{d-1}\beta_1^j\cdot\left[\widetilde{V}_{j+i+1} - \widetilde{V}_i\right]$ is monotonically decreasing from positive to negative. In other words, the forward difference is monotonically decreasing, such that there exists $j$, $1 \leq j \leq d$ and $\lim_{nd\to\infty} k(g_{nd+1})$ is the maximum among all net updates. Moreover, it is obvious that $\lim_{nd\to\infty} k(g_{nd+1})$ is the minimum.

Hence, we can draw the conclusion: $\exists 1 \leq j \leq d$, such that

$$\lim_{nd\to\infty} k(C) = \lim_{nd\to\infty} k(g_{nd+1}) < \lim_{nd\to\infty} k(g_{nd+2}) < \cdots < \lim_{nd\to\infty} k(g_{nd+j})$$

and

$$\lim_{nd\to\infty} k(g_{nd+j}) > \lim_{nd\to\infty} k(g_{nd+j+1}) > \cdots > \lim_{nd\to\infty} k(g_{nd+d+1}) = \lim_{nd\to\infty} k(C),$$

where $K(C)$ is the net update factor for gradient $g_i = C$. $\qquad\square$

## G   PROOF OF LEMMA 4

**Lemma 7.** [1] For a bounded random variable $X$ and a differentiable function $f(x)$, the expectation of $f(X)$ is as follows:

$$\mathbb{E}[f(X)] = f(\mathbb{E}[X]) + \frac{f''(\mathbb{E}[X])}{2}D(X) + R_3 \tag{23}$$

---

[1] See detial in: https://stats.stackexchange.com/questions/5782/variance-of-a-function-of-one-random-variable

where $D(X)$ is variance of $X$, and $R_3$ is as follows:

$$R_3 = \frac{f^{[3]}(\alpha)}{3}\mathbb{E}(X - \mathbb{E}[X])^3 + \tag{24}$$

$$+ \int_{|x-\mathbb{E}[X]|>c} \left( f(\mathbb{E}[X]) + f^{'}(\mathbb{E}[X])(x - \mathbb{E}[X])^2 + f(X) \right) dF(x) \tag{25}$$

$F(x)$ is the distribution function of $X$. $R_3$ is a small quantity under some condition. And $c$ is large enough, such that: for any $\epsilon > 0$,

$$P(X \in [\mathbb{E}[X] - c, \mathbb{E}[X] + c]) = P(|X - \mathbb{E}[X]| \le c) \le 1 - \epsilon \tag{26}$$

*Proof.* (**Proof of Lemma 4** ) In the stochastic online optimization problem equation 7, the gradient subjects the distribution as:

$$g_i = \begin{cases} C, & \text{with probability } p := \frac{1+\delta}{C+1}; \\ -1, & \text{with probability } 1 - p := \frac{C-\delta}{C+1} . \end{cases} \tag{27}$$

Then we can get the expectation of $g_i$ :

$$\mathbb{E}[g_i] = \delta \tag{28}$$

$$\mathbb{E}[g_i^2] = C^2 \cdot \frac{1+\delta}{C+1} + \frac{C-\delta}{C+1} = C + \delta(C+1) \tag{29}$$

$$D[g_i] = C + \delta(C+1) - \delta^2 \tag{30}$$

$$\mathbb{E}[g_i^4] = C(C^2 - C + 1) + \delta(C-1)(C^2+1) \tag{31}$$

$$D[g_i^2] = C^3 - 2C^2 + C + \delta(C^3 - 3C^2 - C - 1) - \delta^2(C+1)^2 \tag{32}$$

Meanwhile, under the assumption that gradients are i.i.d., the expectation and variance of $v_i$ are as following when $nd \to \infty$:

$$\mathbb{E}[v_i] = \lim_{i \to \infty}(1 - \beta_2)\sum_{j=1}^{i}\beta_2^{i-j}\mathbb{E}[g_j^2] = \lim_{i \to \infty}(1 - \beta_2^i)\mathbb{E}[g_j^2] = C + \delta(C+1) \tag{33}$$

$$D[v_i] = \lim_{i \to \infty}(1 - \beta_2)\sum_{j=1}^{i}\beta_2^{i-j}D[g_j^2] = \lim_{i \to \infty}(1 - \beta_2^i)D[g_j^2] = D[g_j^2] \tag{34}$$

Then, for the gradient $g_i$, the net update factor is as follows:

$$k(g_i) = \sum_{t=0}^{\infty} \frac{(1 - \beta_1)\beta_1^t}{\sqrt{\beta_2^{t+1}v_{i-1} + (1 - \beta_2)\beta_2^t \cdot g_i^2 + (1 - \beta_2)\sum_{j=1}^{t}\beta_2^{t-j}g_{i+j}^2}}$$

It should to be clarified that we define $\sum_{j=1}^{t}\beta_2^{t-j}g_{i+j}^2$ equal to zero when $t = 0$. Then we define $X_t$ as:

$$X_t = \beta_2^{t+1}v_{i-1} + (1 - \beta_2)\beta_2^t \cdot g_i^2 + (1 - \beta_2)\sum_{j=1}^{t}\beta_2^{t-j}g_{i+j}^2$$

$$\mathbb{E}[X_t] = \beta_2^{t+1}\mathbb{E}[v_{i-1}] + (1 - \beta_2)\beta_2^t \cdot g_i^2 + (1 - \beta_2)\sum_{j=1}^{t}\beta_2^{t-j}\mathbb{E}[g_{i+j}^2] \tag{35}$$

$$= \beta_2^{t+1}\mathbb{E}[g^2] + (1 - \beta_2)\beta_2^t \cdot g_i^2 + (1 - \beta_2^t)\mathbb{E}[g^2] \tag{36}$$

$$= (1 + \beta_2^{t+1} - \beta_2^t)\mathbb{E}[g^2] + (1 - \beta_2)\beta_2^t \cdot g_i^2 \tag{37}$$

$$D[X_t] = \beta_2^{2(t+1)}D[v_{i-1}] + \frac{(1 - \beta_2)^2(1 - \beta_2^{2t})}{1 - \beta_2^2}D[g_{i+j}^2] \tag{38}$$

$$= \left[ \beta_2^{2(t+1)} + \frac{(1 - \beta_2)^2(1 - \beta_2^{2t})}{1 - \beta_2^2} \right] D[g^2] \tag{39}$$

For the function $f(x) = \frac{1}{\sqrt{x}}$:

$$f''(x) = \frac{3 \cdot x^{-5/2}}{8}$$

According to lemma 7, we can the expectation of $f(X_t)$ as follows:

$$\mathbb{E}[f(X_t)] = (\mathbb{E}[X_t])^{-1/2} + \frac{3}{8}(\mathbb{E}[X_t])^{-5/2} \cdot D[X_t] \tag{40}$$

$\mathbb{E}[X_t]$ and $D[X_t]$ are expressed by equation 35 and equation 38. Then we can obtain the expectation expression of net update factor as follows:

$$k(g_i) = \sum_{t=0}^{\infty}(1-\beta_1)\beta_1^t \left[ \frac{1}{\sqrt{(1-\beta_2)\beta_2^t g_i^2 + (1+\beta_2^{t+1}-\beta_2^k)\mathbb{E}[g_i^2]}} + \frac{3D_t}{8[(1-\beta_2)\beta_2^t g_i^2 + (1+\beta_2^{t+1}-\beta_2^t)\mathbb{E}[g_i^2]]^{\frac{5}{2}}} \right] \tag{41}$$

where $D_t = D[X_k]$. Then for gradient $C$ and $-1$, the net update factor is as follows:

$$k(C) = \sum_{t=0}^{\infty}(1-\beta_1)\beta_1^t \left[ \frac{1}{\sqrt{(1-\beta_2)\beta_2^t C^2 + (1+\beta_2^{t+1}-\beta_2^k)\mathbb{E}[g_i^2]}} + \frac{3D_t}{8[(1-\beta_2)\beta_2^t C^2 + (1+\beta_2^{t+1}-\beta_2^t)\mathbb{E}[g_i^2]]^{\frac{5}{2}}} \right] \tag{42}$$

and

$$k(-1) = \sum_{t=0}^{\infty}(1-\beta_1)\beta_1^t \left[ \frac{1}{\sqrt{(1-\beta_2)\beta_2^t + (1+\beta_2^{t+1}-\beta_2^k)\mathbb{E}[g_i^2]}} + \frac{3D_t}{8[(1-\beta_2)\beta_2^t + (1+\beta_2^{t+1}-\beta_2^t)\mathbb{E}[g_i^2]]^{\frac{5}{2}}} \right] \tag{43}$$

We can see that each term in the infinite series of $k(C)$ is smaller than the corresponding one in $k(-1)$. Thus, $k(C) < k(-1)$.

$\square$

## H    PROOF OF LEMMA 6

*Proof.* From Lemma 2, we can get:

$$\lim_{nd\to\infty} \frac{m_{nd+i}}{\sqrt{v_{nd+i}}} = \frac{\frac{1-\beta_1}{1-\beta_1^d}(C+1)\beta_1^{i-1} - 1}{\sqrt{\frac{1-\beta_2}{1-\beta_2^d}(C^2-1)\beta_2^{i-1} + 1}}$$

We sum up all updates in an epoch, and define the summation as $\mathcal{S}(\beta_1, \beta_2, C)$.

$$\mathcal{S}(\beta_1, \beta_2, C) = \sum_{i=1}^{d} \lim_{nd\to\infty} \frac{m_{nd+i}}{\sqrt{v_{nd+i}}}$$

Assume $\beta_2$ and $C$ are large enough such that $v_t \gg 1$, we get the approximation of limit of $v_{nd+i}$ as:

$$\lim_{nd\to\infty} v_{nd+i} \approx \frac{1-\beta_2}{1-\beta_2^d}(C^2-1)\beta_2^{i-1}$$

Then we can draw the expression of $\mathcal{S}(\beta_1, \beta_2, C)$ as:

$$\begin{aligned}
\mathcal{S}(\beta_1, \beta_2, C) &= \sum_{i=1}^{d} \frac{\frac{1-\beta_1}{1-\beta_2^d}(C+1)\beta_1^{i-1} - 1}{\sqrt{\frac{1-\beta_2}{1-\beta_2^d}(C^2-1)\beta_2^{i-1}}} \\
&= \sum_{i=1}^{d} \frac{\frac{1-\beta_1}{1-\beta_1^d}(C+1)\beta_1^{i-1}}{\sqrt{\frac{1-\beta_2}{1-\beta_2^d}(C^2-1)\beta_2^{i-1}}} - \sum_{i=1}^{d} \frac{1}{\sqrt{\frac{1-\beta_2}{1-\beta_2^d}(C^2-1)\beta_2^{i-1}}} \\
&= \sqrt{\frac{1-\beta_2^d}{(1-\beta_2)\beta_2^{d-1}}}\sqrt{\frac{C+1}{C-1}} \cdot \frac{1-\beta_1}{1-\beta_1^d} \cdot \frac{\sqrt{\beta_2^d}-\beta_1^d}{\sqrt{\beta_2}-\beta_1} - \sqrt{\frac{1-\beta_2^d}{(1-\beta_2)\beta_2^{d-1}}} \cdot \frac{1}{\sqrt{C^2-1}} \frac{\sqrt{\beta_2^d}-1}{\sqrt{\beta_2}-1} \\
&= \sqrt{\frac{1-\beta_2^d}{(1-\beta_2)\beta_2^{d-1}(C-1)}} \left[ \frac{(1-\beta_1)(\beta_2^d-\beta_1^d)\sqrt{C+1}}{(1-\beta_1^d)(\sqrt{\beta_2}-\beta_1)} - \frac{\sqrt{\beta_2^d}-1}{\sqrt{C+1}(\sqrt{\beta_2}-1)} \right]
\end{aligned}$$

Let $\mathcal{S}(\beta_1, \beta_2, C) = 0$, we get the equation about critical condition:

$$C + 1 = \frac{(1 - \beta_1^d)(\sqrt{\beta_2^d} - \beta_1^d)(1 - \sqrt{\beta_2})}{(1 - \beta_1)(\sqrt{\beta_2} - \beta_1)(1 - \sqrt{\beta_2^d})}$$

$\square$

## I  HYPER-PARAMETERS INVESTIGATION

### I.1  HYPER-PARAMETERS SETTING

Here, we list all hyper-parameter setting of all above experiments.

Table 5: Hyper-parameter setting of logistic regression in Figure 2.

| Optimizer | learning rate | $\beta_1$ | $\beta_2$ | $n$ |
|---|---|---|---|---|
| SGD | 0.1 | N/A | N/A | N/A |
| Adam | 0.001 | 0 | 0.999 | N/A |
| AMSGrad | 0.001 | 0 | 0.999 | N/A |
| non-AdaShift | 0.001 | 0 | 0.999 | 1 |
| max-AdaShift | 0.01 | 0 | 0.999 | 1 |

Table 6: Hyper-parameter setting of Multilayer Perceptron on MNIST in Figure 3.

| Optimizer | learning rate | $\beta_1$ | $\beta_2$ | $n$ |
|---|---|---|---|---|
| SGD | 0.001 | N/A | N/A | N/A |
| Adam | 0.001 | 0 | 0.999 | N/A |
| AMSGrad | 0.001 | 0 | 0.999 | N/A |
| non-AdaShift | 0.0005 | 0 | 0.999 | 1 |
| max-AdaShift | 0.01 | 0 | 0.999 | 1 |

Table 7: Hyper-parameter setting of WGAN-GP in Figure 7a.

| Optimizer | learning rate | $\beta_1$ | $\beta_2$ | $n$ |
|---|---|---|---|---|
| Adam | 1e-5 | 0 | 0.999 | N/A |
| AMSGrad | 1e-5 | 0 | 0.999 | N/A |
| AdaShift | 1.5e-4 | 0 | 0.999 | 1 |

Table 8: Hyper-parameter setting of Neural Machine Translation BLEU in Figure 7b.

| Optimizer | learning rate | $\beta_1$ | $\beta_2$ | $n$ |
|---|---|---|---|---|
| Adam | 0.0001 | 0.9 | 0.999 | N/A |
| AMSGrad | 0.0001 | 0.9 | 0.999 | N/A |
| AdaShift | 0.01 | 0.9 | 0.999 | 30 |

Table 9: Hyper-parameter setting of ResNet on Cifar-10 in Figure 4, DenseNet on Cifar-10 in Figure 5 and DenseNet on Tiny-Imagenet in Figure 6.

| Optimizer | learning rate | $\beta_1$ | $\beta_2$ | $n$ |
|---|---|---|---|---|
| Adam | 0.001 | 0.9 | 0.999 | N/A |
| AMSGrad | 0.001 | 0.9 | 0.999 | N/A |
| AdaShift | 0.01 | 0.9 | 0.999 | 10 |

## I.2    LEARNING RATE $\alpha_t$ SENSITIVITY

In this section, we discuss the learning rate $\alpha_t$ sensitivity of AdaShift. We set $\alpha_t \in \{0.1, 0.01, 0.001\}$ and let $n = 10$, $\beta_1 = 0.9$ and $\beta_2 = 0.999$. The results are shown in Figure 9 and Figure 10. Empirically, we found that when using the $max$ spatial operation, the best learning rate for AdaShift is around ten times of Adam.

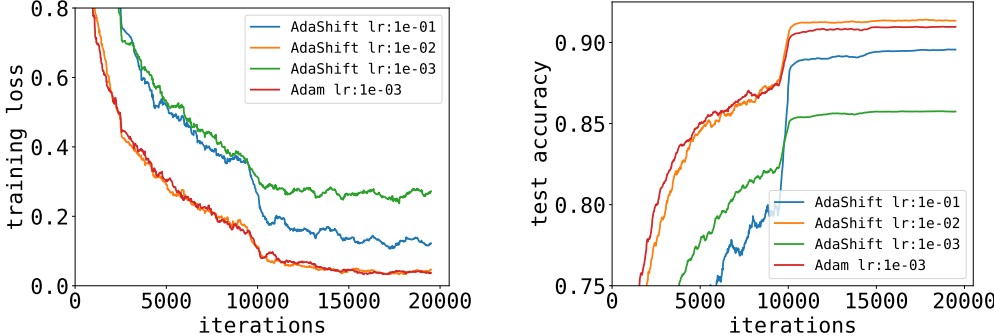

Figure 9: Learning rate sensitivity experiment with ResNet on CIFAR-10.

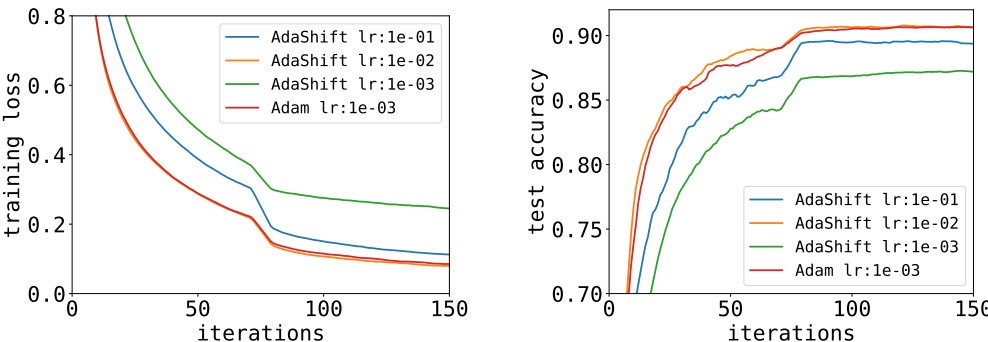

Figure 10: Learning rate sensitivity experiment with DenseNet on CIFAR-10.

## I.3    $\beta_1$ AND $\beta_2$ SENSITIVITY

In this section, we discuss the $\beta_1$ and $\beta_2$ sensitivity of AdaShift. We set $\alpha = 0.01$, $n = 10$ and let $\beta_1 \in \{0, 0.9\}$ and $\beta_2 \in \{0.9, 0.99, 0.999\}$. The results are shown in Figure 11 and Figure 12. According to the results, AdaShift holds a low sensitivity to $\beta_1$ and $\beta_2$. In some tasks, using the first moment estimation (with $\beta_1 = 0.9$ and $n = 10$) or using a large $\beta_2$, e.g., 0.999 can attain better performance. The suggested parameters setting is $n = 10, \beta_1 = 0.9, \beta_2 = 0.999$.

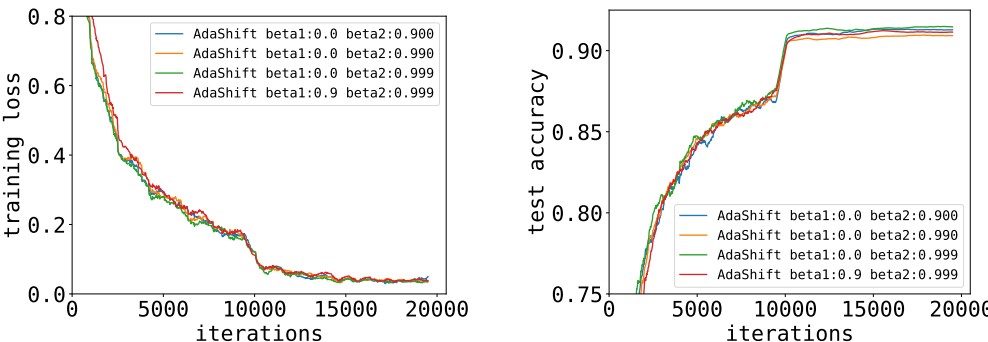

Figure 11: $\beta_1$ and $\beta_2$ sensitivity experiment with ResNet on CIFAR-10.

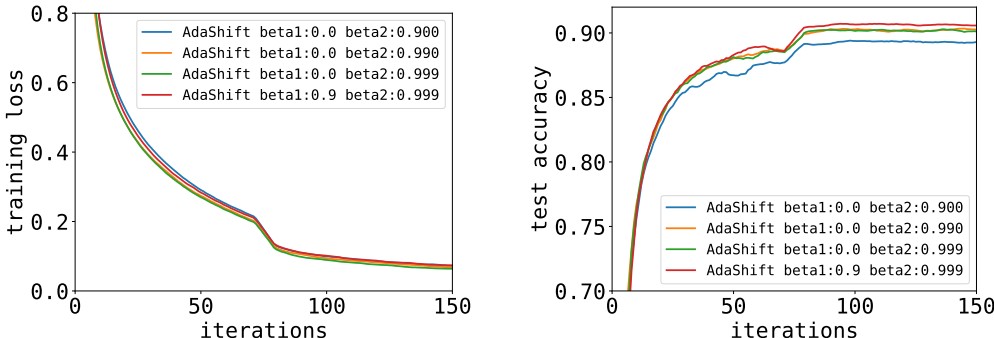

Figure 12: $\beta_1$ and $\beta_2$ sensitivity experiment with DenseNet on CIFAR-10.

### I.4  $n$ AND $m$ SENSITIVITY

In this section, we discuss the $n$ sensitivity of AdaShift. Here we also test a extended version of first moment estimation where it only uses the latest $m$ gradients ($m \leq n$):

$$v_t = \beta_2 v_{t-1} + (1 - \beta_2)\phi(g_{t-n}^2) \ \text{ and } \ m_t = \frac{\sum_{i=0}^{m-1} \beta_1^i g_{t-i}}{\sum_{i=0}^{m-1} \beta_1^i}. \tag{44}$$

We set $\beta_1 = 0.9$, $\beta_2 = 0.999$. The results are shown in Figure 13, Figure 14 and Figure 15. In these experiments, AdaShift is fairly stable when changing $n$ and $m$. We have not find a clear pattern on the performance change with respect to $n$ and $m$.

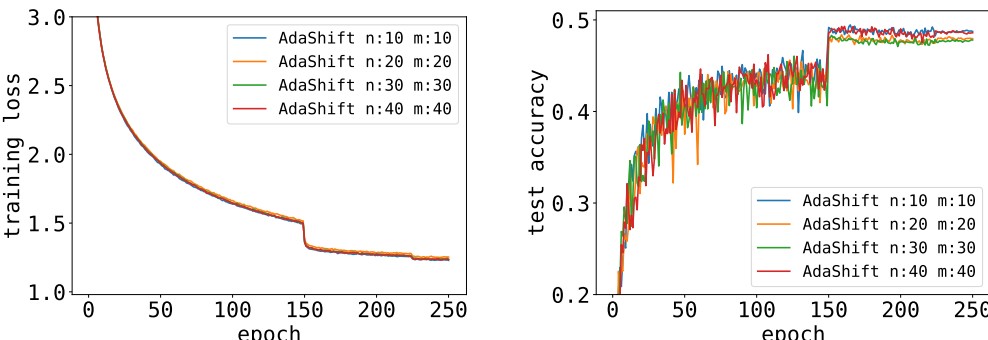

Figure 13: $n$ sensitivity experiment with DenseNet on Tiny-ImageNet.

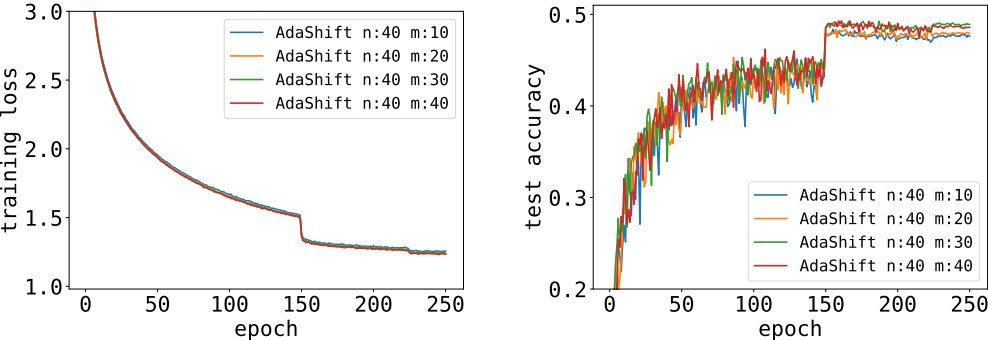

Figure 14: $m$ sensitivity experiment with DenseNet on Tiny-ImageNet.

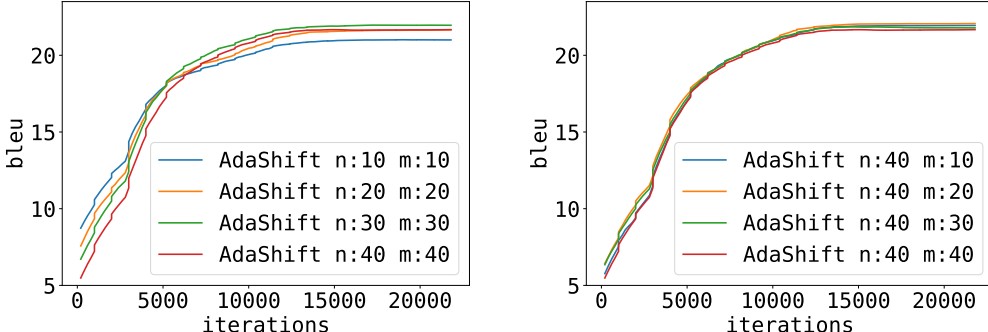

Figure 15: $n$ and $m$ sensitivity experiment with Neural Machine Translation BLEU.

## J  TEMPORAL-ONLY AND SPATIAL-ONLY

In our proposed algorithm, we apply a spatial operation on the temporally shifted gradient $g_{t-n}$ to update $v_t$: $v_t[i] = \beta_2 v_{t-1}[i] + (1 - \beta_2)\phi(g_{t-n}^2[i])$. It is based on the temporal independent assumption, i.e., $g_{t-n}$ is independent of $g_t$. And according to our argument in Section 4.2, one can further assume every element in $g_{t-n}$ is independent of the $i$-th dimension of $g_t$.

We purposely avoid involving the spatial elements of the current gradient $g_t$, where the independence might not holds: when a sample which is rare and has a large gradient appear in the mini-batch $x_t$, the overall scale of gradient $g_t$ might increase. However, for the temporally already decorrelation $g_{t-i}$, further taking the advantage of the spatial irrelevance will not suffer from this problem.

We here provide extended experiments on two variants of AdaShift: (i) AdaShift (temporal-only), which only uses the vanilla temporal independent assumption and evaluate $v_t$ with: $v_t = \beta_2 v_{t-1} + (1 - \beta_2)g_{t-n}^2$; (ii) AdaShift (spatial-only), which directly uses the spatial elements without temporal shifting.

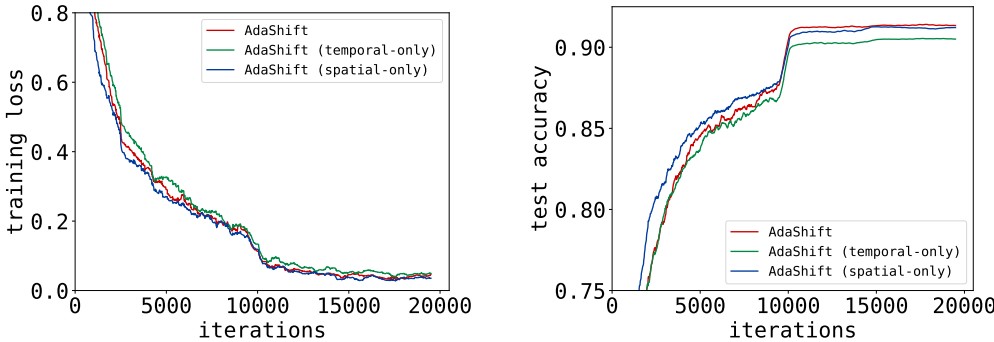

Figure 16: ResNet on CIFAR-10.

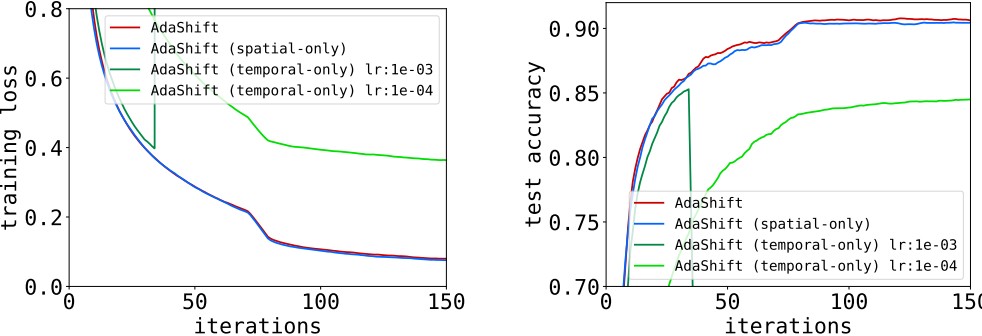

Figure 17: DenseNet on CIFAR-10.

According to our experiments, AdaShift (temporal-only), i.e., without the spatial operation, is less stable than AdaShift. In some tasks, AdaShift (temporal-only) works just fine; while in some other cases, AdaShift (temporal-only) suffers from explosive gradient and requires a relatively small learning rate. The performance of AdaShift (spatial-only) is close to Adam. More experiments for AdaShift (spatial-only) are included in the next section.

# K EXTENDED EXPERIMENTS: NADAM AND ADASHIFT(SPACE ONLY)

In this section, we extend the experiments and add the comparisons with Nadam and AdaShift (spatial-only). The results are shown in Figure 18, Figure19 and Figure20. According to these experiments, Nadam and AdaShift (spatial-only) share similar performance as Adam.

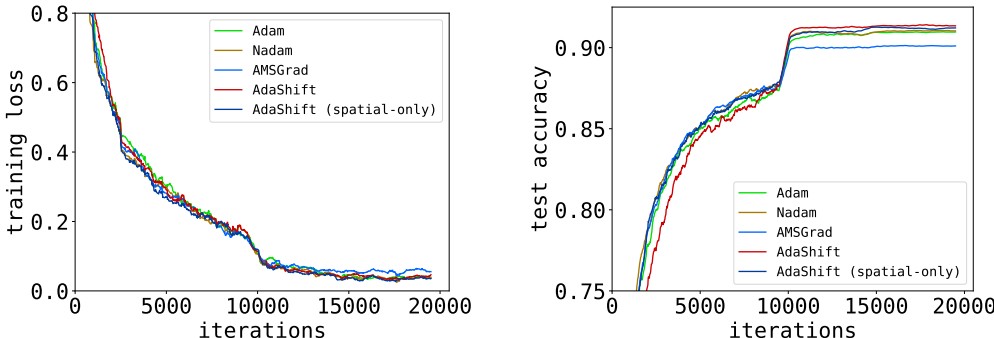

Figure 18: ResNet on CIFAR-10.

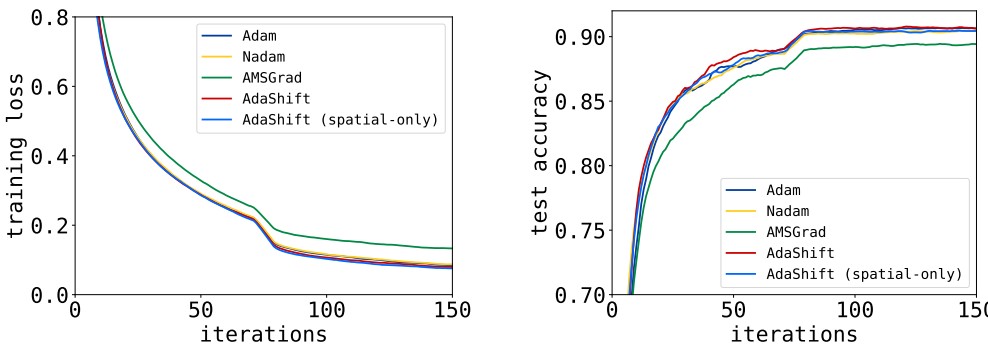

Figure 19: DenseNet on CIFAR-10.

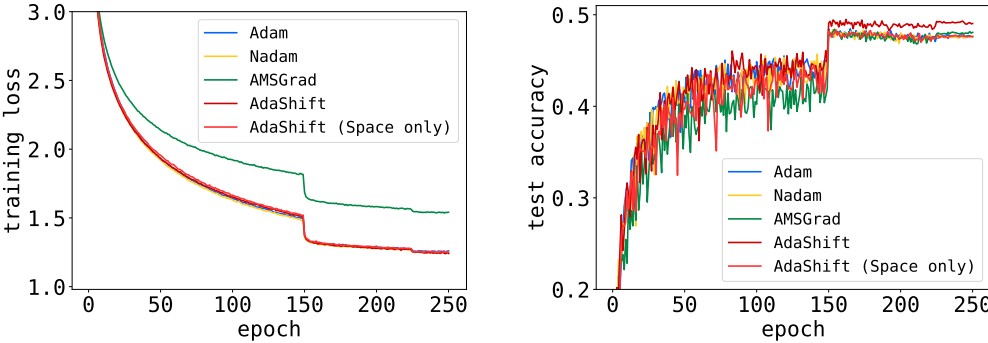

Figure 20: DenseNet on Tiny-ImageNet.

## L    EXTENSION EXPERIMENTS: ILL-CONDITIONED QUADRATIC PROBLEM

Rahimi & Recht raise the point, at test of time talk at NIPS 2017, that it is suspicious that gradient descent (aka back-propagation) is ultimate solution for optimization. A ill-conditioned quadratic problem with Two Layer Linear Net is showed to be challenging for gradient descent based methods, while alternative solutions, e.g., Levenberg-Marquardt, may converge faster and better. The problem is defined as follows:

$$L(W_1, W_2; A) = \mathbb{E}_{x \in N(0,1)} \|W_1 W_2 x - Ax\|^2 \tag{45}$$

where $A$ is some known badly conditioned matrix ($k = 10^{20}$ or $10^5$), and $W_1$ and $W_2$ are the trainable parameters.

We test SGD, Adam and AdaShift with this problem, the results are shown in Figure 21, Figure 24. It turns out as long as the training goes enough long, SGD, Adam, AdaShift all basically converge in this problem. Though SGD is significantly better than Adam and AdaShift.

We would tend to believe this is a general issue of adaptive learning rate method when comparing with vanilla SGD. Because these adaptive learning rate methods generally are scale-invariance, i.e., the step-size in terms of $g_t/sqrt(v_t)$ is basically around one, which makes it hard to converge very well in such a ill-conditioning quadratic problem. SGD, in contrast, has a step-size $g_t$; as the training converges SGD would have a decreasing step-size, makes it much easier to converge better. The above analysis is confirmed with Figure 22 and Figure 23, with a decreasing learning rate, Adam and AdaShfit both converge very good.

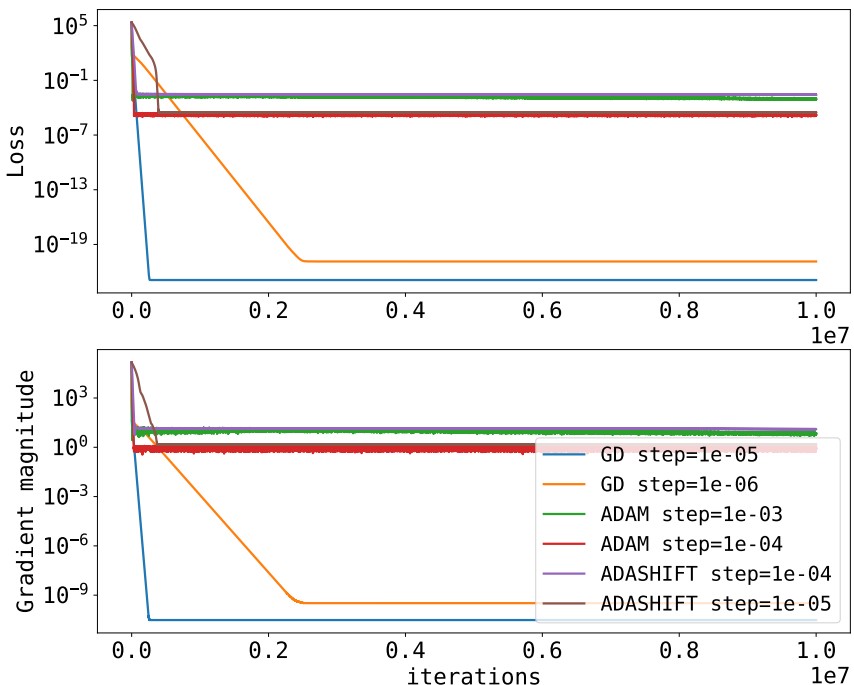

Figure 21: Ill-conditioned quadratic problem, with fixed learning rate.

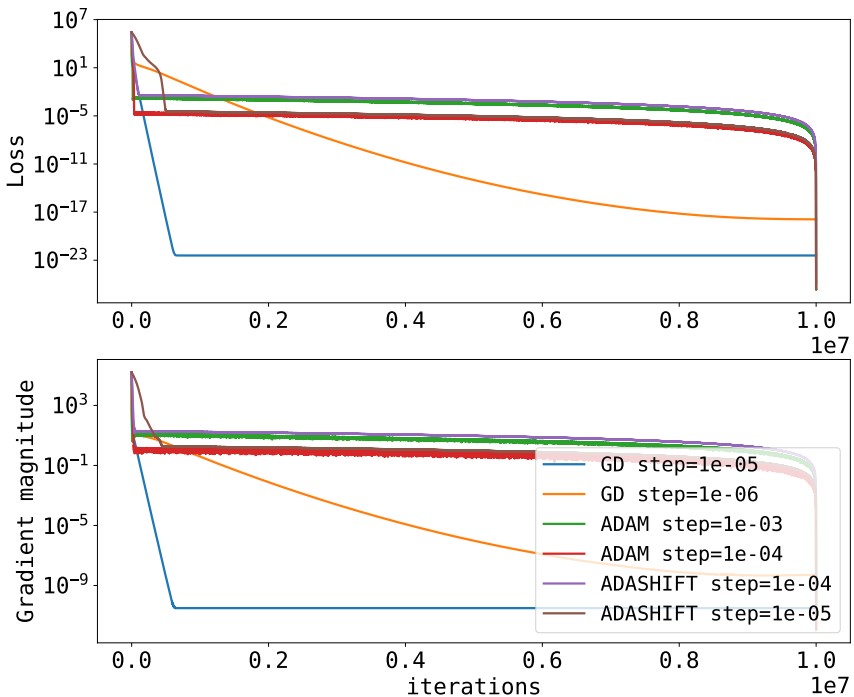

Figure 22: Ill-conditioned quadratic problem, with linear learning rate decay.

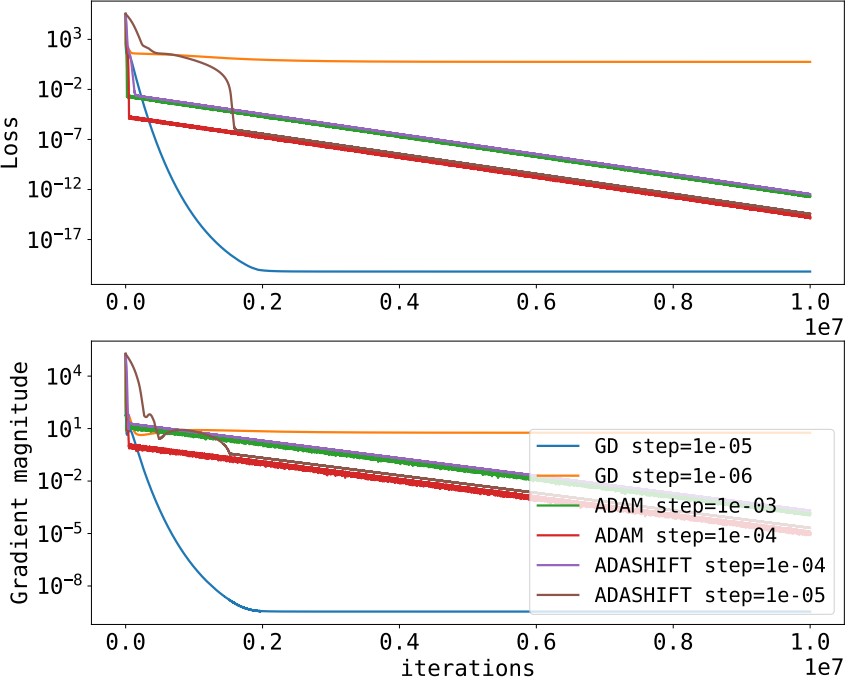

Figure 23: Ill-conditioned quadratic problem, with exp learning rate decay.

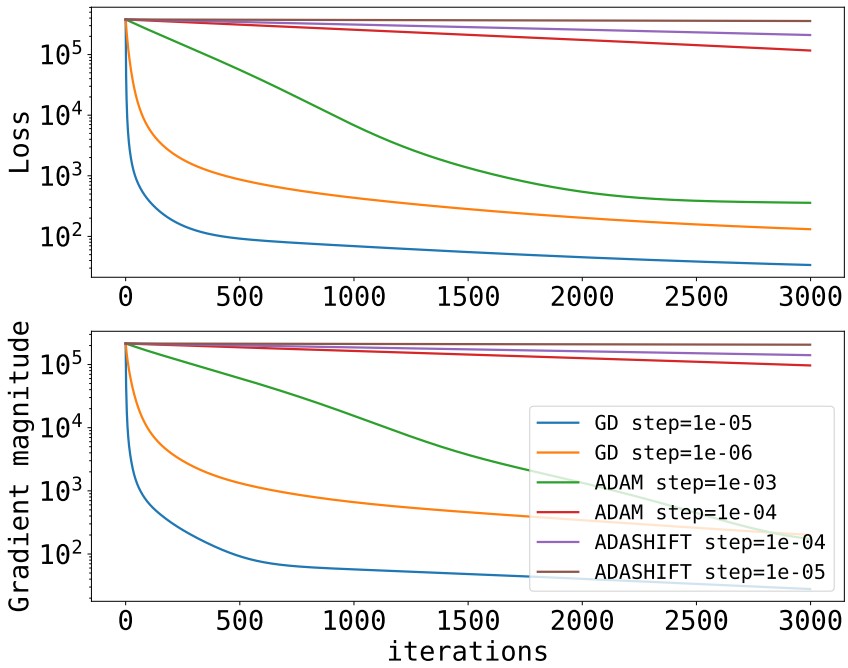

Figure 24: Ill-conditioned quadratic problem, with fixed learning rate and insufficient iterations.

