# OpenReview forum: "AdaShift: Decorrelation and Convergence of Adaptive Learning Rate Methods"
_ICLR.cc/2019/Conference_

### Official Review · AnonReviewer2 · 2018-10-31
**Analyses and fixes one problem of ADAM that could be specific or general**

**Rating:** 9
**Confidence:** 4

**Review:**

This manuscript contributes a new online gradient descent algorithm with adaptation to local curvature, in the style of the Adam optimizer, ie with a diagonal reweighting of the gradient that serves as an adaptive step size. First the authors identify a limitation of Adam: the adaptive step size decreases with the gradient magnitude. The paper is well written.

The strengths of the paper are a interesting theoretical analysis of convergence difficulties in ADAM, a proposal for an improvement, and nice empirical results that shows good benefits. In my eyes, the limitations of the paper are that the example studied is a bit contrived and as a results, I am not sure how general the improvements.

# Specific comments and suggestions

Under the ambitious term "theorem", the results of theorem 2 and 3 limited to the example of failure given in eq 6. I would have been more humble, and called such analyses "lemma". Similarly, theorem 4 is an extension of this example to stochastic online settings. More generally, I am worried that the theoretical results and the intuitions backing the improvements are built only on one pathological example. Are there arguments to claim that this example is a prototype for a more general behavior?


Ali Rahimi presented a very simple example of poor perform of the Adam optimizer in his test-of-time award speech at NIPS this year (https://www.youtube.com/watch?v=Qi1Yry33TQE): a very ill-conditioned factorized linear model (product of two matrices that correspond to two different layers) with a square loss. It seems like an excellent test for any optimizer that tries to be robust to ill-conditioning (as with Adam), though I suspect that the problem solved here is a different one than the problem raised by Rahimi's example.


With regards to the solution proposed, temporal decorrelation, I wonder how it interacts with mini-batch side. With only a light understanding of the problem, it seems to me that large mini-batches will decrease the variance of the gradient estimates and hence increase the correlation of successive samples, breaking the assumptions of the method.


Using a shared scalar across the multiple dimensions implies that the direction of the step is now the same as that of the gradient. This is a strong departure compared to ADAM. It would be interesting to illustrate the two behaviors to optimize an ill-conditioned quadratic function, for which the gradient direction is not a very good choice.


The performance gain compared to ADAM seems consistent. It would have been interesting to see Nadam in the comparisons.



I would like to congratulate the authors for sharing code.

There is a typo on the y label of figure 4 right.

---

> ### Author Response · Authors · 2018-11-18
> **Response to Reviewer 2**
>
> Thanks for your constructive feedback.
>
> Q: In my eyes, the limitations of the paper are that the example studied is a bit contrived and as a result, I am not sure how general the improvements. More generally, I am worried that the theoretical results and the intuitions backing the improvements are built only on one pathological example. Are there arguments to claim that this example is a prototype for a more general behavior?
>
> >> We mixed the general arguments for the non-convergence of Adam into these analyses of counterexamples. According to the reviewers' feedback, we realize that it is indeed confusing. We thus have reorganized the analysis section, and clearly separated the analysis on counterexamples and the general arguments on the non-convergence issue of Adam. Actually, ‘‘assigning relatively small step-size to large gradient and assigning relatively large step-size to small gradient’’ is the general behavior of Adam and traditional adaptive learning rate methods. Sometimes it causes non-convergence, and more generally, it just hampers the convergence. Please see the reorganized arguments in Section 3.3 for details.
>
> Q: With regards to the solution proposed, temporal decorrelation, I wonder how it interacts with the mini-batch side. With only a light understanding of the problem, it seems to me that large mini-batches will decrease the variance of the gradient estimates and hence increase the correlation of successive samples, breaking the assumptions of the method.
>
> >> The argument is thought-provoking. But it seems that, though decreasing the variance makes the difference between samples smaller, it does not change the independence. Assume that the gradients are independently sampled from a standard Gaussian N(0, 1). If the Gaussian is squeezed to N(0, 0.1), gradients sampled from the squeezed Gaussian are still independent of each other. Using our argument in the paper, we still reach the same conclusion: assuming the loss function is fixed, as long as these mini-batches are independently sampled, no matter the mini-batch size is large or small, their gradients are always independent.
>
> Q: The performance gain compared to Adam seems consistent. It would have been interesting to see Nadam in the comparisons.
>
> >> We have conducted a set of experiments for Nadam. The results are presented in Appendix K. Generally, we found Nadam shows quite similar performance as Adam. Please check Appendix K for details.
>
> Q: Ali Rahimi presented a very simple example of the poor performance of the Adam optimizer in his test-of-time award speech at NIPS this year. It seems like an excellent test for any optimizer that tries to be robust to ill-conditioning (as with Adam), though I suspect that the problem solved here is a different one than the problem raised by Rahimi's example.
>
> >> It is an interesting test and we have tested our algorithm with the code they provided. Our finding is somewhat weird: as long as the training is sufficiently long, SGD, Adam, and AdaShift basically converge in this problem, though the final performance of SGD is significantly better than Adam and AdaShift.
>
> >> We tend to believe this is a general issue of adaptive learning rate method when comparing with vanilla SGD. Because these adaptive learning rate methods are generally scale-invariance, i.e., the step-size in terms of g_t/sqrt(v_t) is basically around 1, which makes it hard to converge very well in such an ill-conditioning quadratic problem. SGD, in contrast, has a step-size g_t. As the training converges, SGD would have a decreasing step-size, making it much easier to converge better. To confirm our analysis, we train the same task with a decreasing learning rate, and we found that at the end of the training, Adam and AdaShfit both converge satisfactorily.
>
> >> Levenberg-Marquardt, which minimizes $(\delta W_1, \delta W_2)$ by solving least-squares, shows the fastest convergence. It indicates the possibility of better alternatives to gradient descent (backpropagation) based optimization, which deserves further investigations.

---

### Official Review · AnonReviewer3 · 2018-11-01
**Another fix of non-convergence of Adam -- AdaShift**

**Rating:** 6
**Confidence:** 4

**Review:**

In this paper, the authors found that decorrelating $v_t$ and $g_t$ fixes the non-convergence issue of Adam. Motivated by that, AdaShift that uses a temporal decorrelation technique is proposed. Empirical results demonstrate the superior performance of AdaShift compared to Adam and AMSGrad. My detailed comments are listed as below.

1) Theorem 2-4 provides interesting insights on Adam. However, the obtained theoretical results rely on specific toy problems (6) and (13). In the paper, the authors mentioned that "... apply the net update factor to study the behaviors of Adam using Equation 6 as an example. The argument will be extended to the stochastic online optimization problem and general cases." What did authors mean the general cases?

2) The order of presenting Algorithm 1, 2 and Eq. (17) should be changed. I suggest to first present AdaShift (i.e., Eq. (17) or Algorithm 3 with both modified adaptive learning rate and moving average), and then elaborate on temporal decorrelation and others. AdaShift should be presented as a new Algorithm 1.  In experiments, is there any result associated with the current Algorithm 1 and 2? If no, why not compare in experiments? One can think that Algorithm 1 and 2 are adaptive learning rate methods against adaptive gradient methods (e.g., Adam, AMSGrad).

3) Is there any convergence rate analysis of AdamShift even in the convex setting?

4) The empirical performance of AdamShift is impressive. Can authors mention more details on how to set the hyperparameters for AdamShift, AMSGrad, Adam, e.g., learning rate, \beta 1, and \beta 2?

---

> ### Author Response · Authors · 2018-11-18
> **Response to Reviewer 3**
>
> Thanks for your constructive feedback.
>
> Q: However, the obtained theoretical results rely on specific toy problems (6) and (13). In the paper, the authors mentioned that "... apply the net update factor to study the behaviors of Adam using Equation 6 as an example. The argument will be extended to the stochastic online optimization problem and general cases." What did the authors mean the general cases?
>
> >> We are sorry for the confusion. We mixed the general arguments and the counterexample-specific arguments together. According to the reviewers’ feedback, we have reorganized the analysis section, and now the analysis on counterexamples and the general arguments on the non-convergence of Adam are separated. We would appreciate if you could have a check on these reorganized arguments (Section 3.3). The general arguments are actually very sound.
>
> Q: The empirical performance of AdaShift is impressive. Can authors mention more details on how to set the hyperparameters for AdaShift, AMSGrad, Adam, e.g., learning rate, \beta 1, and \beta 2?
>
> >> In the revision, we have listed hyperparameter settings in each experiment in Appendix. We have also conducted a set of experiments on hyperparameter sensitivities of AdaShift, which are also included in Appendix. Please check these details in Appendix I of the new version of our paper.
>
> Q: I suggest to first present AdaShift (i.e., Eq. (17) or Algorithm 3 with both modified adaptive learning rate and moving average), and then elaborate on temporal decorrelation and others. AdaShift should be presented as a new Algorithm 1.
>
> >> Thanks a lot for this valuable suggestion. We have tried your suggestion and it looks much better. Please check it in the revised version.
>
> Q: Is there any convergence rate analysis of AdaShift even in the convex setting?
>
> >> Currently, we do not have convergence rate analysis for AdaShift. We will work on it and hope it will appear soon.

---

> > ### Comment · AnonReviewer3 · 2018-11-26
> > **An improved version**
> >
> > I think the authors have further improved the paper, thus I have increased my score to 6.
> >
> > However, some further theoretical analysis should be made in the future work.
> >
> > "We are sorry for the confusion. We mixed the general arguments and the counterexample-specific arguments together. According to the reviewers’ feedback, we have reorganized the analysis section, and now the analysis on counterexamples and the general arguments on the non-convergence of Adam are separated. We would appreciate if you could have a check on these reorganized arguments (Section 3.3). The general arguments are actually very sound. "
> >
> > The current Section 3.3  provides general claims only based on the common clues from specific counterexamples.
> > Thus, it is important to give convergence analysis of AdaShift in the general case. Although this part is missing in the current version, I am fine with it due to its novel idea. I list two references which provide the convergence analysis of Adam-type algorithms for nonconvex optimization. I hope these can help authors to build their own analysis.
> >
> >
> > Zhou, Dongruo, et al. "On the convergence of adaptive gradient methods for nonconvex optimization." arXiv preprint arXiv:1808.05671 (2018).
> >
> > Chen, Xiangyi, et al. "On the convergence of a class of adam-type algorithms for non-convex optimization." arXiv preprint arXiv:1808.02941 (2018).

---

> > > ### Author Response · Authors · 2018-11-27
> > > **Thank you so much.**
> > >
> > > Thank you so much for the comments and useful references. We will put our effort into the convergence analysis. Hopefully, we will have some convergence analysis in our final version.

---

### Official Review · AnonReviewer1 · 2018-11-02
**Original contribution to stochastic optimizers, with presentation to be rearranged**

**Rating:** 6
**Confidence:** 4

**Review:**

Summary
------

Based on an extensive argument acoordig to which Adam potential failures are due to the positive correlation between gradient and moment estimation, the authors propose Adashift, a method in which temporal shift (and more surprisingly 'spatial' shift, ie mixing of parameters) is used to ensure that moment estimation is less correlated with gradient, ensuring convergence of Adashift in pathological cases, without the efficiency cost of simpler method such as AMSGrad. An extensive analysis of a pathological counter example, introduced in Reddi et al. 2018 is analysed, before the algorithm presentation and experimental validation. Experiments shows that the algorithm has equivalent speed as Adam and sometimes false local minima, resulting in better training error, and potentially better test error.

Review
-------

The decorrelation idea is original and well motivated by an extensive analysis of a pathological examples. The experimental validation is thorough and convincing, and the paper is overall well written.

Regarding content, the reviewer is quite dubious about the spatial decorrelation idea. ASsuming shared moment estimation for blocks of parameters is definitely meaningful from an information perspective, and has indeed been used before, but it seems to have little to do with the 'decorrelation' idea. The reviewer would be curious to see a comparison with temporal-only adashift in the experiment, as the block / max operator \phi, to isolate the temporal and 'spatial' effect.

Regarding presentation, the reviewer's opinion is that the paper is too long. Too much space is spent discussing an interesting yet limited counterexample, on which 5 theorems (that are simple analytical derivations) are stated. This should be summarized (and its interesting argument stated more concisely), to the benefit of the actual algorithm presentation, that should appear in the main text (algorithm 3). The spatial decorrelation method, that remains unclear to the reviewer, should be discussed more and validated more extensively. The current size of the paper is 10 pages, which is much above the ICLR average length.

However, due to the novelty of the algorithm, the reviewer is in favor of accepting the paper, provided the authors can address the comments above.

---

> ### Author Response · Authors · 2018-11-18
> **Response to Reviewer 1**
>
> Thanks for your constructive feedback.
>
> Q: Regarding content, the reviewer is quite dubious about the spatial decorrelation idea. Assuming shared moment estimation for blocks of parameters is definitely meaningful from an information perspective, and has indeed been used before, but it seems to have little to do with the 'decorrelation' idea.
>
> >> In our proposed algorithm, only the spatial elements of temporally-shifted gradient g_{t-n} are involved in the calculation of v_t. Based on the temporal independence assumption, g_{t-n} is independent of g_t, which naturally implies that all elements in g_{t-n} are independent of the elements in g_t. Thus, using the spatial elements in g_{t-n} does not break the independence assumption. We have revised the related sections and avoided the term ‘‘spatial independence’’ that is indeed confusing.
>
> Q: Regarding presentation, the reviewer's opinion is that the paper is too long. Too much space is spent discussing an interesting yet limited counterexample, on which 5 theorems (that are simple analytical derivations) are stated. This should be summarized (and its interesting argument stated more concisely), to the benefit of the actual algorithm presentation, that should appear in the main text (Algorithm 3). The spatial decorrelation method, that remains unclear to the reviewer, should be discussed more and validated more extensively. The current size of the paper is 10 pages, which is much above the ICLR average length.
>
> >> Thanks a lot for these constructive suggestions. We have rewritten related sections accordingly. The main changes are: (i) we have renamed the analytical derivations as lemmas and removed unnecessary details; (ii) we have reorganized the analysis section to make it more concise and clear; (iii) we have removed Algorithms 1 and 2, and directly presented Algorithm 3; (iv) we have made the arguments on the validity of using spatial elements much more clear.
>
> Q: The reviewer would be curious to see a comparison with temporal-only AdaShift in the experiment, as the block/max operator \phi, to isolate the temporal and 'spatial' effect.
>
> >> We have added experiments on temporal-only AdaShift and spatial-only AdaShift. Some experiments on temporal-only AdaShift can be found in Figure 2 and Figure 3 in the experiments Section, and more results are included in Appendix J and K.
>
> >> Temporal-only AdaShift is actually not as stable as AdaShift. It works well in simple tasks, but it suffers from explosive gradient in complex systems: a neuron recovering from a vanishing gradient state is the typical failure case, where v_t is nearly zero. AdaShift with spatial operation, in contrast, does not suffer from this problem: the gradients of an entire block is relatively stable and won’t vanish.
>
> >> Spatial-only AdaShift turns out not to fit our assumption, but it is indeed a very interesting extension of Adam. Therefore, we have also conducted a set of experiments on spatial-only AdaShift. According to our initial investigations, ‘‘spatial-only AdaShift’’ shares a similar performance to Adam. Details are presented in Appendix J and K.

---

### Public Comment · (anonymous) · 2018-10-02
**There is nothing in the code you provided**

You claim that "The anonymous code is provided at http://bit.ly/2NDXX6x", but there is nothing there.
It has been almost a week since the submission was closed. Do you plan to upload the code some days later but before the official reviewers start reading your paper? I don't like this behavior.

Seriously, it should be regarded as some kinds of "cheating". You can successfully pretend that you had done everything before the submission if no one notices that. Reviewers may think that you've done more than others. It is unfair to other authors that honestly admit they haven't managed/refactored the code yet.

I don't think whether you publish the code in the review period can strongly affect the result of acceptance. Honestly admitting that you haven't made your code really is much better than "cheating".

---

> ### Author Response · Authors · 2018-10-04
> **The code is now accessible.**
>
> Thanks for your interest in our paper and sorry for not releasing the code in time. The code is now accessible from the provided link.
>
> We think publicizing the code should be done before the review process, rather than after paper acceptance. And from our perspective, releasing the code bears no relation to contribution, but the authors'  duty.

---

> > ### Public Comment · (anonymous) · 2018-10-09
> > **Thanks**
> >
> > Thank you for your kind reply! I think my tone might be too serious before. Your paper is good and I just want to say you don't need that kind of little "tricks". :) Hope you can have a good result.

---

### Public Comment · ~Mikhail_Konobeev1 · 2019-01-07
**ICLR Reproducibility Challenge**

We reproduced some of the experiments from this paper as a part of the ICLR reproducibility challenge. While we replicated the optimizer and the experiments in PyTorch, we would like to thank the authors for sharing their implementation in TensorFlow as that made verification of the results easier. Overall, our findings are close to the results from the paper and we summarize them as follows:

* While the authors have not experimented with the sequential online optimization problem proposed by Reddi et al. (2018) since it did not fit their assumptions, we found that the proposed AdaShift optimizer is able to converge but is slower than AMSGrad. At the same time we also confirm the results of the experiment with a stochastic counterexample on which Adam fails.

* In the logistic regression and MLP experiments on MNIST the results of all optimizers are quite similar in our experiments as well as in the paper.

* We also tried to reproduce more complex experiments such as WGAN-GP and NMT. In WGAN-GP we note the atypical way in which the authors conducted their experiments with a generative model, specifically, by fixing generator and training only discriminator. We trained both discriminator and generator and in our experiments using AdaShift lead to worse Inception Score than using either Adam or AMSGrad which in turn had quite similar results. Additionally, we note that the authors set a penalty coefficient to 0.1 which is significantly different from its value in the paper that proposed this method where it was equal to 10. We also note that authors use maxGP version of the loss function in their code. In our opinion, motivation and discussion of these choices would be beneficial. At the same time for the translation model, AdaShift showed the best result which is similar to the author’s experiment and the difference from other methods is even more significant.

* We found a discrepancy between the implementation of the AdaShift provided by the authors and its description in the paper: in the paper it is repeatedly stated that the result of applying the function \phi to the gradient should be squared, while in the code the function is applied to the square of the gradient. Could you please elaborate on this?  While we found that on the synthetic and MNIST experiments both orders lead to very similar results, we think that clarification is still needed.

---

> ### Author Response · Authors · 2019-01-08
> **Thanks a lot!**
>
> Thanks a lot for your interest in our paper and your reproduction of our results.
>
> * GANs is currently a mystery. When you joint training the generator and discriminator, there exists a lot of issues on the minimax conflict and generator-discriminator balance/tradeoff in GANs, even using the relatively mature WGAN. So, we choose to train the discriminator only, i.e.., fixing the generator.
>
> * WGAN requires the discriminative function to be k-Lipschitz. However, how to enforce the Lipschitz is currently a not well-addressed problem. Though WGAN-GP provides a solution which is well accepted, we believe it is not an accurate implementation of Lipschitz. We thus choose MaxGP which we believe is a more accurate implementation. See Appendix E in [1] for details.
>
> * The penalty coefficient is set to 0.1 because we empirically found that no matter for MaxGP or the original GP, it is better than 10.0 in most cases. So, we use 0.1 as the default.
>
> * Thanks for point out the discrepancy between the implementation and the description in the paper. We think we have miswritten it in the paper, though in the current form of the paper, to match the implementation, we only need reformulation the function phi to be sqrt(max x^2). We choose the square the gradient first because we'd like $v$ to reflect the gradient scale, otherwise, the max operation will ignore the negative ones. We will revise the paper to make it more clear. Thanks.
>
> [1] Understanding the Effectiveness of Lipschitz-Continuity in Generative Adversarial Nets: https://openreview.net/pdf?id=r1zOg309tX
>
> BTW: did you try different learning rate for AdaShift in the WGAN experiments? Its best learning rate is usually around ten times the learning rate of Adam. According to our experiences, they should be comparable.

---

> > ### Public Comment · ~Mikhail_Konobeev1 · 2019-01-14
> > **WGAN experiment**
> >
> > Dear authors,
> > We agree that when training GANs a lot of issues may arise such as generator discriminator disbalance, the requirement of a discriminator to be k-Lipschitz (in the case of WGAN). However, at this point we were not able to achieve results with AdaShift that are comparable to or better than the results with Adam or AMSGrad and given that this is an interesting problem that is much more realistic than training a generator against a fixed discriminator that could be used to study the behavior of an optimizer, we find this experiment more useful. We note that training a discriminator against a fixed generator essentially leads to training a neural network to distinguish dataset images from noise which is a peculiar setting. We conducted experiments in two settings when training both discriminator and generator. First is the typical setting WGAN-GP with penalty coefficient equal to 10 and the second is WGAN max-GP with penalty coefficient equal to 0.1. While in the first setting we were able to train the generative model with all optimizers (albeit with AdaShift performing worse than both Adam and AMSGrad), we found that in the second setting the results are much worse for all optimizers and the generative model does not train well (we were not able to achieve Inception Score of at least 4 after 100 epochs of training). In our case making learning rate for AdaShift 10 times bigger than the learning rates of Adam and AMSGrad lead to divergence in the first setting. We also tried a number of other learning rates and again were not able to achieve results for AdaShift that are comparable to the other optimizers in the first setting. We would greatly appreciate it if you could provide further details that may lead to comparable results between different optimizers.

---

> > > ### Author Response · Authors · 2019-01-15
> > > **A quick reply.**
> > >
> > > In our experiments, we have also found that GP sometimes leads to training divergence, we suggest using MaxGP, which we have found usually can solve this problem.
> > >
> > > Fixing discriminator and training generator is a well-defined optimization problem, but it is less interesting we think. Because it is merely finding the optimal x, the x that holds the maximum D(x). Or more strictly, optimizing all x towards the local optimal. This is kind of simulating the typical cause of mode collapse problem in GANs.
> > >
> > > Given fixed P_r and P_g and training the discriminator is estimating the given distance metric, say Wasserstein distance, between P_r and P_g, which is a sound optimization problem we believe. And we suspect that being hard to get the optimal discriminative function is one important cause of why GAN is currently not easy to train. This is also what motivates the authors to study optimizer.
> > >
> > > If the memory is not wrong, we have achieved Inception Score around 8.0 with AdaShift (comparable with Adam). For more experiment details, we will post another response in a few weeks.

---

> ### Author Response · Authors · 2019-04-23
> **Code of AdaShift for GANs training.**
>
> Hi, we have published our code of AdaShift on GANs.
>
> See https://github.com/ZhimingZhou/AdaShift-Lipschitz-GANs-MaxGP.
>
> ------------
>
> This repo includes the implementation of AdaShift and also the demonstration code that uses AdaShift to training GANs which achieves FID: 15.8800±0.4921 and Inception Score: 8.0367±0.0499 for unsupervised image generation of GANs in CIFAR-10.
>
> The provided implementation of AdaShift (common/optimizer/AdaShift) is further developed version, which extends our discussion in [1], i.e., v_t can be any random variable that keeps the scale of the gradients and is independent of g_t. We use LGANs developed in [2] and MaxGP described in [3].
>
> [1] AdaShift: Decorrelation and Convergence of Adaptive Learning Rate Methods https://arxiv.org/abs/1810.00143
>
> [2] Lipschitz Generative Adversarial Nets https://arxiv.org/abs/1902.05687
>
> [3] Towards Efficient and Unbiased Implementation of Lipschitz Continuity in GANs https://arxiv.org/abs/1904.01184
>
> We use tensorflow 1.5 with python 3.5. You can refer to setting_cuda9_cudnn7_tensorflow1.5.sh to build up your environment. Try the code via running: python3 realdata_resnet.py. synthetic_real.py and synthetic_toy.py are the code we used for the synthetic experiments in [2] and [3].

---

### Meta-Review · Area_Chair1 · 2018-12-14

**Confidence:** 3
**Recommendation:** Accept (Poster)

**Metareview:**

This paper proposes a new stochastic optimization scheme similar to Adam. The authors claim that Adam can be improved upon by decorrelating the second-moment estimate v_t from gradient estimates g_t. This is done through the temporal decorrelation scheme, as well as block-wise sharing of estimates v_t.

The reviewers agree that the paper is sufficiently well-written, original and significant to be accepted for ICLR, although some unclarity remains after the reviews. A disadvantage of the method is mainly an increased computational cost (linear in 'n', however this might be negligible when sharing v_t across blocks).